# Architecture of symbiotic dinoflagellate photosystem I–light-harvesting supercomplex in *Symbiodinium*

Long-Sheng Zhao[1,2,3,7], Ning Wang[1,7], Kang Li [3,7], Chun-Yang Li [1,3,7], Jian-Ping Guo[4], Fei-Yu He[2], Gui-Ming Liu[5], Xiu-Lan Chen [2,3], Jun Gao [4] ✉, Lu-Ning Liu [1,6] ✉ & Yu-Zhong Zhang [1,2,3] ✉

*Symbiodinium* are the photosynthetic endosymbionts for corals and play a vital role in supplying their coral hosts with photosynthetic products, forming the nutritional foundation for high-yield coral reef ecosystems. Here, we determine the cryo-electron microscopy structure of *Symbiodinium* photosystem I (PSI) supercomplex with a PSI core composed of 13 subunits including 2 previously unidentified subunits, PsaT and PsaU, as well as 13 peridinin-Chl *a*/*c*-binding light-harvesting antenna proteins (AcpPCIs). The PSI–AcpPCI supercomplex exhibits distinctive structural features compared to their red lineage counterparts, including extended termini of PsaD/E/I/J/L/M/R and AcpPCI-1/3/5/7/8/11 subunits, conformational changes in the surface loops of PsaA and PsaB subunits, facilitating the association between the PSI core and peripheral antennae. Structural analysis and computational calculation of excitation energy transfer rates unravel specific pigment networks in *Symbiodinium* PSI–AcpPCI for efficient excitation energy transfer. Overall, this study provides a structural basis for deciphering the mechanisms governing light harvesting and energy transfer in *Symbiodinium* PSI–AcpPCI supercomplexes adapted to their symbiotic ecosystem, as well as insights into the evolutionary diversity of PSI–LHCI among various photosynthetic organisms.

Photosystem I (PSI) is a multi-subunit pigment-protein supercomplex that plays a crucial role in the light-driven photosynthetic electron transport of oxygenic photosynthesis. It accepts electrons from the Photosystem II (PSII)-mediated water oxidation delivered by plastocyanin or cytochrome $c_6$ and transfers electrons to ferredoxin[1]. In eukaryotes, the PSI supercomplex consists of a core complex and peripheral light-harvesting complexes (LHCs). The LHCs associated with the PSI core (LHCI) capture and transfer light energy to the core complex, which is composed of the reaction center P700 and core antennae[1]. The structures of core subunits are highly conserved across cyanobacteria, algae, and plants[2–15]. In contrast, the LHCIs show high variability in their number, protein composition, and association of

[1]MOE Key Laboratory of Evolution and Marine Biodiversity, Frontiers Science Center for Deep Ocean Multispheres and Earth System & College of Marine Life Sciences, Ocean University of China, Qingdao 266003, China. [2]Marine Biotechnology Research Center, State Key Laboratory of Microbial Technology, Shandong University, Qingdao 266237, China. [3]Laboratory for Marine Biology and Biotechnology, Laoshan Laboratory, Qingdao 266237, China. [4]Hubei Key Laboratory of Agricultural Bioinformatics, College of Informatics, Huazhong Agricultural University, Wuhan 430070, China. [5]Beijing Key Laboratory of Agricultural Genetic Resources and Biotechnology, Institute of Biotechnology, Beijing Academy of Agriculture and Forestry Sciences, 100097 Beijing, China. [6]Institute of Systems, Molecular and Integrative Biology, University of Liverpool, Liverpool L69 7ZB, UK. [7]These authors contributed equally: Long-Sheng Zhao, Ning Wang, Kang Li, Chun-Yang Li. ✉e-mail: gaojun@mail.hzau.edu.cn; luning.liu@liverpool.ac.uk; zhangyz@sdu.edu.cn

pigments among different oxygenic photoautotrophs, representing the intrinsic evolutionary and regulatory solutions utilized by photo-autotrophic organisms to thrive in varying environmental niches[16].

Dinoflagellates are an ecologically and economically important group of unicellular eukaryotes, serving as primary producers widespread in various aquatic environments[17–19]. The plastids of the vast majority of photosynthetic dinoflagellates originated from a primordial endosymbiont[20,21]. One typical paradigm of dinoflagellate symbiosis is found in marine dinoflagellates belonging to the genus *Symbiodinium*. These photosynthetic endosymbionts form a crucial relationship with reef-building corals (zooxanthellae) in subtropical and tropical shallows, creating the trophic foundation for one of the world's most diverse and productive marine ecosystems, namely coral reefs[22–24]. Such symbiotic systems exhibit remarkable productivity despite in nutrient-poor waters in which they thrive, attributed to the intimate mutualism between corals and photosynthetic *Symbiodinium*, which support the living of one-quarter to one-third of all marine species[25].

The symbiotic relationship between *Symbiodinium* and corals is complex and delicate, and their mutualism is susceptible to environmental factors including temperature, light, and salinity[26]. The environmental stresses, such as high temperature and high light in the shallow ocean, could result in photoinhibition, pigment degradation, and oxidative damage in *Symbiodinium*, hampering nutrient flow between *Symbiodinium* and corals and consequently, leading to the devastating death of coral colonies[27,28]. Previous studies have

suggested the role of *Symbiodinium* PSI–LHCI in photoprotection[29,30]. However, how *Symbiodinium* evolves the PSI–LHCI supercomplex to fulfill efficient energy and electron transfer, charge separation, and photoprotection remains poorly understood, and the lack of high-resolution structure has precluded a better understanding of *Symbiodinium* PSI–LHCI structure and function to ensure cell fitness in specific ecological environments.

Here, we solved the structure of PSI–LHCI from *Symbiodinium* using single-particle cryo-electron microscopy (cryo-EM) at a resolution of 2.8 Å. Our study reveals distinct architectural features and excitation energy transfer pathways of *Symbiodinium* PSI–LHCI, providing insights into the mechanisms of light energy harvesting and transfer of the PSI–LHCI. These features may be essential for *Symbiodinium* cell fitness in specific symbiotic environments they form with other organisms.

## Results and discussion
### Overall structure
The PSI–LHCI supercomplex was purified from *Symbiodinium* sp. (Clade A) (LeadingTec, Shanghai, China) grown in a free-living state and the heavier green band was collected and analyzed (Supplementary Fig. 1, Supplementary Data 1). Using single-particle cryo-EM, we solved the PSI–LHCI supercomplex structure at an overall resolution of 2.8 Å (Supplementary Fig. 2, Supplementary Table 1). *Symbiodinium* PSI–LHCI supercomplex comprises a PSI core complex and a peripheral antenna system containing 13 peridinin-Chl *a/c*-binding light-harvesting proteins

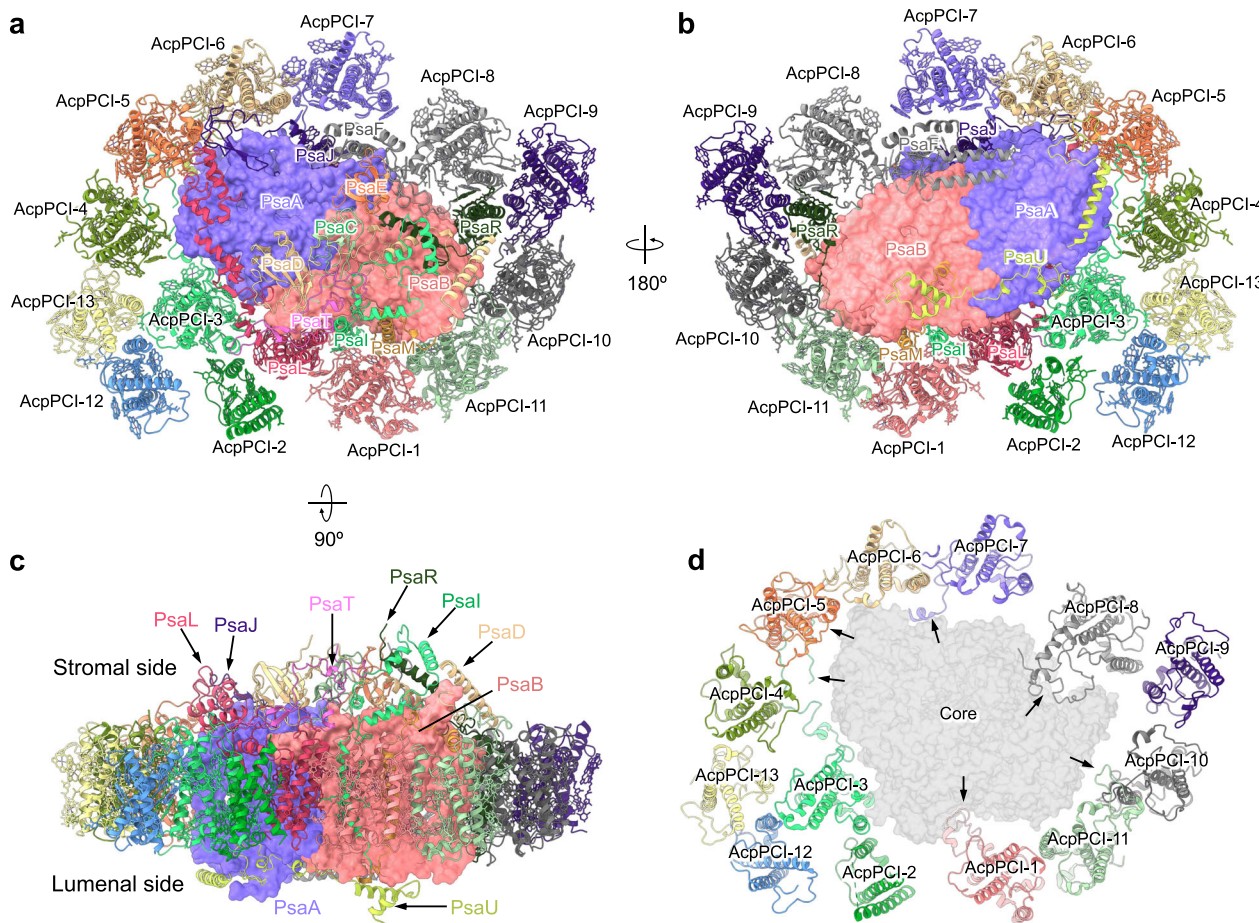

**Fig. 1 | Overall structure of the *Symbiodinium* PSI–AcpPCI supercomplex.** **a** PSI–AcpPCI supercomplex viewed from stromal side. Core subunits are labeled as their names. LHCs are labeled as AcpPCI. Previously unidentified PSI core subunit is named as "PsaT". **b** PSI–AcpPCI supercomplex viewed from lumenal side.

Previously unidentified PSI core subunit is named as "PsaU". **c** Side view of the PSI–LHC supercomplex. **d** AcpPCIs in PSI–LHC supercomplex. The long termini of AcpPCIs are indicated by arrows.

(AcpPCs)[31] that form a ring around the core (Fig. 1). Hereafter, the AcpPCs that are associated with PSI were named as AcpPCI. The PSI core is composed of 13 subunits, including PsaA–F, PsaI, PsaJ, PsaL, PsaM, and PsaR, as well as 2 previously unidentified subunits, PsaT (NCBI GenBank: PP196340) at the stromal surface and PsaU (NCBI GenBank: PP196339) at the lumenal surface (Fig. 1a–c, Supplementary Fig. 3a). To identify the amino acid sequences of PsaT and PsaU, small regions of the density maps of PsaT and PsaU with higher resolution were selected (Supplementary Fig. 3b, e), and amino acid sequences were manually speculated based on the selected map (Speculation sequence) (Supplementary Fig. 3c, f). Then, blastp against the transcriptome sequences was performed using the speculation sequences to identify potential sequences (Supplementary Fig. 3c, f). Each of the potential sequences was manually fitted into the selected map, and amino acids with distinguishable side chains (such as phenylalanine, tyrosine, tryptophan, arginine, and glycine) were used to identify the proper sequence from the candidate sequences (Supplementary Fig. 3b, e). The sequences were further confirmed by the suitability of the side chains for the density map in other regions (Supplementary Fig. 3d, g). Moreover, we assigned 196 Chl $a$, 20 Chl $c$, 41 diadinoxanthin (Ddx), 13 $\beta$-carotene, 16 dinoxanthin (Dx), 10 peridinin (Per), 2 phylloquinones, 3 $Fe_4S_4$ clusters, and 21 lipid molecules in the PSI–AcpPCI structure (Supplementary Table 2). Intriguingly, distinct from other reported PSI–LHCI structures, most of the PSI core subunits and half of the AcpPCI subunits possess extended terminal structures that span across both stromal and lumenal surfaces of the supercomplex (Fig. 1c, Supplementary Figs. 3–5). This specific organization, together with the membrane-protruded PsaT and PsaU subunits, enhances the longitudinal volume and inter-subunit interactions of the supercomplex, potentially promoting the structural integrity and functionality of *Symbiodinium* PSI–AcpPCI in symbiotic environment.

## Structural features of the *Symbiodinium* PSI core

The *Symbiodinium* PSI core exhibits several specific structural properties compared to other reported PSI core complexes. Like diatom PSI, the *Symbiodinium* PSI lacks PsaK and PsaO but possesses the PsaR subunit, indicating its close evolutionary relationship with diatom PSI (Supplementary Fig. 6, Supplementary Table 3)[3,4,13,14]. In contrast to the relatively conserved transmembrane (TM) domains, notable variations are observed in the extrinsic loop structures of PsaA and PsaB and the terminal domains of the PsaD/E/I/J/L/M/R subunits (Supplementary Figs. 4, 5, 7).

The extended termini of PsaI/J/L/M/R, along with the PsaT subunit, form a protective layer that encompasses the stromal surface of the PSI core (Supplementary Fig. 8), resulting in reduced interactions between the reaction center and molecules in the stroma. This may mitigate the damage of reactive oxygen species to the reaction center and protect the pigments in the reaction center under high light. Moreover, these extended termini effectively create a specific PsaC/D/E-ferredoxin domain, presumably facilitating the binding of ferredoxin to PSI (Supplementary Fig. 8a–d). On the lumenal surface, the PsaU subunit spans across the edges of PsaA and PsaB without obstructing the binding sites for cytochrome $c_6$ or plastocyanin. Such distinct structural features may enable efficient interactions and intermolecular electron transfer between PSI and electron carriers (Supplementary Fig. 8e). Intriguingly, these long terminal extensions exhibit homologous sequences exclusive to dinoflagellate species, specifically in *Symbiodinium* (Supplementary Fig. 9). Likewise, the homologous sequences of PsaT and PsaU are exclusive in *Symbiodiniaceae*, with a high degree of conservation. These distinguishing characteristics provide evidence for a more diversified structure of the PSI core than was previously recognized.

The extended N-terminal domains of PsaL and PsaJ are situated on the stromal surface of PsaA, in proximity to the interface of PsaA and AcpPCIs (Fig. 2a). Intriguingly, both N-termini interact with each other, forming a molecular "thread" that stitches the stromal loops of AcpPCI-3, AcpPCI-5, and AcpPCI-6 together to strengthen the association of AcpPCI antennae (Fig. 2a, Supplementary Fig. 10). Moreover, their binding also leads to the conformational changes of PsaA at the N-terminal loop as well as the I2-V6, Q44-S47, Q164-S171, C276-D294, and Q362-I368 loops.

Interactions among the extended C-termini of PsaD and PsaI and the N-terminus of PsaR lead to formation of a "cap-like" structure on top of the stromal surface of PsaB (Supplementary Fig. 8). The C-terminus of PsaD directly binds to PsaB, extending across its surface and inducing the conformational changes of the PsaB V358-S366 and I148-S180 loops (Fig. 2b, Supplementary Fig. 11a). The PsaD C-terminus further extends to the outer edge of PsaB, forming interactions with AcpPCI-10 and AcpPCI-11, which may promote the physical association and energy transfer between AcpPCIs and the PSI core (Supplementary Fig. 11b). PsaD also has a longer N-terminal loop than those of reported PSI structures, forming interactions with the L628-V633 loop of PsaB (Supplementary Figs. 9, 11c). The extended termini of PsaI and PsaR interact with each other and form intense hydrogen bonds with adjacent subunits, thereby stabilizing their conformations (Fig. 2c, e; Supplementary Fig. 11d–f). The PsaI C-terminal extension may also adapt to a shortened N-terminus of PsaB (Fig. 2c, Supplementary Figs. 7, 11d).

The extended C-terminus of PsaM is positioned at the interface of the PSI core and AcpPCIs, and interacts with the I148-S180 and Y261-T270 loops of PsaB, the additional loop of PsaD, as well as adjacent AcpPCI-10 and AcpPCI-11 (Fig. 2d, Supplementary Figs. 5, 12a–c). The extended N-terminus of PsaM interacts with the N-terminus of PsaI and binds to the lumenal surface of PsaB, corresponding to the shortened PsaB C-terminus and T89-A102 loop (Fig. 2d, Supplementary Fig. 12d). The extended termini of PsaE and PsaF form intense interactions with adjacent subunits (Supplementary Fig. 12e, f). The lumenal loops of PsaB (A215-S223 and G429-S449) are shortened, mirroring their counterparts in PsaA (G213-L244 and A421-I431). This prevents interference with the extended N-terminus of PsaF and the lumenal loops of PsaR and AcpPCI-10 (Supplementary Fig. 12g).

The PsaT subunit (NCBI GenBank: PP196340) interacts with multiple core subunits, including PsaB–D, PsaL, and PsaI (Supplementary Fig. 13a, b). These associations may facilitate the stabilization of these extrinsic subunits. Interestingly, the PsaS subunit found in diatom PSI cores occupies a similar position to PsaT[3,4]; however, their structures exhibit low similarity (Fig. 2f). The association of the PsaU subunit (NCBI GenBank: PP196339) results in alternation of the G213-L244, A421-I431, and Y539-C561 loop structures of PsaA (Fig. 2g). Its N-terminus interacts with the extended N-terminus of PsaM, whereas its middle loop and C-terminus interact with AcpPCI-3 and AcpPCI-5/6, respectively, thereby mediating the association of AcpPCIs to the PSI core (Supplementary Fig. 13c).

## Structures of AcpPCI subunits

Despite some similarities in subunit structure and association with PSI with the previously reported LHCs[2–14], *Symbiodinium* AcpPCIs exhibit a notable level of distinctiveness. The binding positions of AcpPCI-6/7 are highly conserved across the LHCs of *Symbiodinium*, red algae, cryptophytes, and diatoms, whereas conformational shifts and rotations occur in the remaining AcpPCIs (Supplementary Fig. 14), owing to the specific core structures and loops of AcpPCIs. For instance, AcpPCI-1 rotates 180 degrees compared to its counterparts in red lineage PSI–LHCIs (Supplementary Fig. 14d, f). Since PsaK and PsaO are absent in the *Symbiodinium* PSI core, AcpPCI-3 and AcpPCI-4 occupy their positions and facilitate the binding of peripheral AcpPCIs, resembling diatom FCPI-3 and FCPI-4[3,4] (Supplementary Fig. 14c, f). However, AcpPCI-3 exhibits a slight shift compared to diatom FCPI-3, while AcpPCI-4 rotates about 90 degrees compared to FCPI-4, resulting in significant changes in the positions and orientations of AcpPCI-

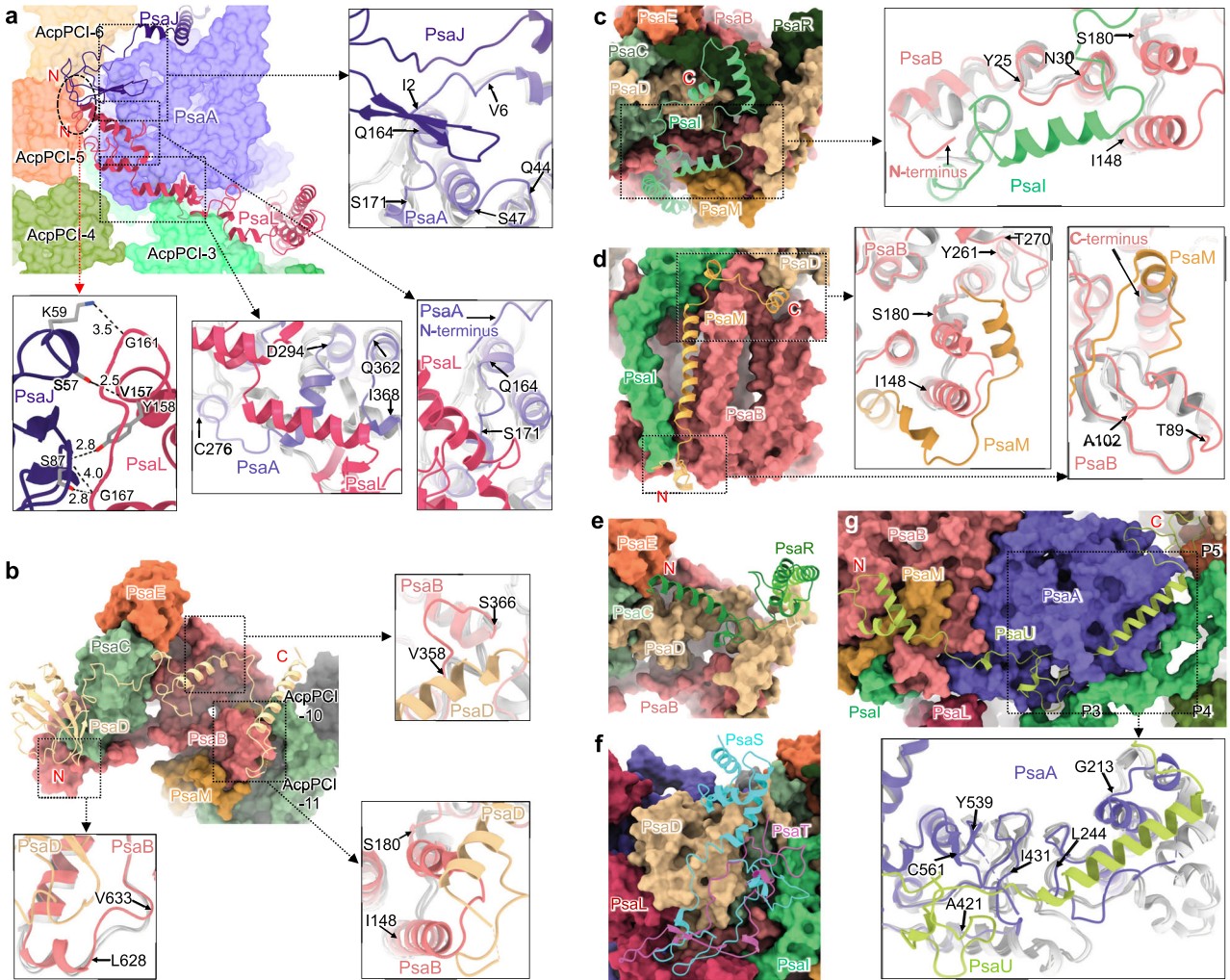

**Fig. 2 | Structures and locations of subunits PsaL, PsaJ, PsaD, PsaI, PsaM, PsaE, PsaR and previously unidentified subunits PsaT and PsaU and their interactions with PsaA and PsaB.** The structures of the extended termini of PsaL/PsaJ (**a**), PsaD (**b**), PsaI (**c**), and PsaM (**d**). Interactions between the extended termini of PsaJ and PsaL (oval) are indicated by red arrow. The boxed areas show the changes of the loop structures of PsaA or PsaB indicated by arrows, facilitating the binding of the extended termini of these subunits. The N-terminus and C-terminus are labeled.

The PsaA and PsaB of red algae (PDB: 7Y5E), cryptophyte (PDB: 7Y7B) and diatom (PDB: 6LY5) are colored gray. **e** The structure of the extended terminus of PsaR. **f** Comparison of the structures and locations of PsaT of *Symbiodinium* and PsaS of diatom. **g** The structures and location of PsaU. The boxed areas show the changes of the loop structures of PsaA indicated by arrows, facilitating the binding of PsaU. The PsaA of red algae, cryptophyte and diatom are colored gray.

2/12/13 (Supplementary Fig. 14e, f). AcpPCI-5/8/9/10/11 shift toward to the PSI core. The specific conformation and arrangement enable the AcpPCIs to tightly associate with the PSI core, facilitating excitation energy flux from AcpPCIs to the PSI core.

The three TM helices (αA, αB, αC) of individual AcpPCIs are relatively conserved. In contrast, their loop structures exhibit flexibility (Fig. 3a–c; Supplementary Figs. 15, 16). AcpPCI-1/3/5/7/8/11 possess additional terminal loops that can interact with the PSI core. Similar extended terminal loops were also found in diatom FCPI-4/21/23/24 and green algal Lhca5/6[3,4,6,8,9], which primarily interact with adjacent LHCs. Phylogenetic analysis indicates that AcpPCIs have a close evolutionary relationship with diatom FCPIs, belonging to the Lhcr (AcpPCI-4/5/6/7/8/10/11) and Lhcf (AcpPCI-1/2/3/9/12/13) families (Supplementary Fig. 17). Lhcr-type AcpPCI-6/7/11 exhibit similar structures and share similarities with their counterparts in red lineage PSI–LHCIs, except for AcpPCI-7 that has an extended C-terminus and AcpPCI-11 with an extended N-terminus (Supplementary Fig. 18a). These extended regions interact with PsaA/J and the extended terminus of PsaD, respectively (Supplementary Figs. 11b, 19a, b). Lhcr-type AcpPCI-4/5/10 have short αB helices and loop structures that differ

from AcpPCI-6/7/11 (Supplementary Fig. 18b). Notably, AcpPCI-5 possesses a longer N-terminus and AcpPCI-10 has a unique AB-loop compared to its counterparts in red lineage PSI–LHCIs, which associate with the extended termini of PsaL/J and the specific loop (A215-S223) of PsaB, respectively (Supplementary Figs. 12g, 18b, 19c, d). Lhcr-type AcpPCI-8 exhibits a high degree of similarity to its diatom counterpart, FCPI-8, except for an extended N-terminal loop that interacts with PsaA/B/E/F and the extended N-terminus of PsaR (Supplementary Figs. 18c, 19e). These interactions contribute to the proper association of AcpPCIs within the PSI–AcpPCI supercomplex.

The Lhcf-type AcpPCI-3 exhibits a similar structure to diatom FCPI-3 but possesses a very long C-terminal loop that is absent in FCPI-3[3,4] (Supplementary Fig. 18d). This extended loop spans along the interface between PsaA and AcpPCI-4/5, filling the gap resulting from the shortening of the loop structures of PsaA (G213-L244 and A421-I431) and enhancing the binding of AcpPCIs (Supplementary Fig. 19f). Lhcf-type AcpPCI-1/2/9/12/13 possess similar structures, including shorter αB helices compared to the Lhcr-type AcpPCIs (Supplementary Fig. 18e). AcpPCI-1, which is rotated by 180 degrees, possesses a longer C-terminal loop compared to its counterparts in red lineage PSI-

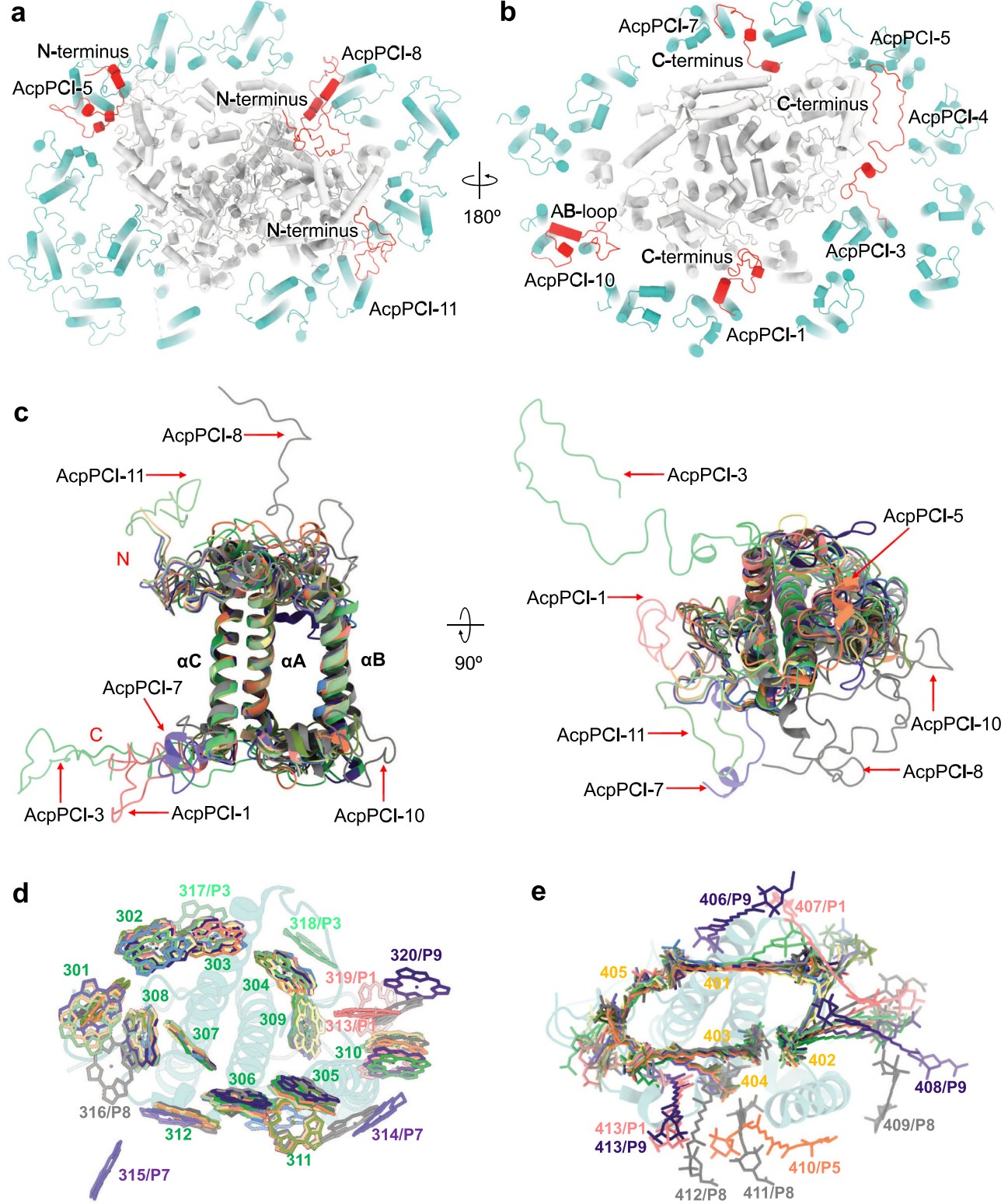

**Fig. 3 | Structures and pigment-binding sites of AcpPCIs.** The structures and arrangements of AcpPCIs (sea green) viewed from stromal side (**a**) and lumenal side (**b**). The long termini of AcpPCIs and AB-loop of AcpPCI-10 are colored red. **c** Structural comparison of 13 AcpPCIs. The long terminal loops of AcpPCIs and AB-loop of AcpPCI-10 are indicated by red arrows. **d** Superposition of the Chl sites of all AcpPCIs viewed from the stromal side. P1/3/7/8/9 represent AcpPCI-1/3/7/8/9. **e** Superposition of the carotenoid sites of all ACPIs viewed from the stromal side. Colors for the AcpPCIs and their pigments in (**c**, **d**, **e**) are in line with those in Fig. 1.

LHCIs[4,13,14]. This longer C-terminal loop extends towards the lumenal side of the PsaI-PsaL region and occupies the position of the specific AB loops found in its counterparts (Supplementary Fig. 14d), providing an anchoring point for AcpPCI-1 (Supplementary Fig. 19g). AcpPCI-9

possesses a shorter αB helix and longer AB and BC loops compared to its Lhcr-type counterparts in red lineage PSI-LHCIs (Supplementary Fig. 18e). These structural differences likely contribute to the conformational shift of AcpPCI-9 towards the PSI core (Supplementary

Fig. 14). Similarly, AcpPCI-13 also exhibits a shorter αB helix compared to diatom FCPI-13 (Supplementary Fig. 18e). The rotations of AcpPCI-12 and AcpPCI-13 relative to FCPI-12 and FCPI-13 result in their short αB helices facing AcpPCI-3, enabling them to occupy less space and position closer to AcpPCI-3 (Supplementary Fig. 14e).

Overall, *Symbiodinium* PSI–AcpPCI shares common architectural features with red algal PSI–LHCR, cryptophyte PSI–ACPI, and diatom PSI–FCPI; on the other hand, it also possesses specific characteristics in the protein structure and organization (Supplementary Fig. 14f). This provides insights into the structural diversity of PSI–LHCI in the red lineage. The unusually extended termini of AcpPCIs and core subunits ensure tight inter-molecular interactions of the PSI–AcpPCI supercomplex and result in the structural variations in the surface loops of PsaA and PsaB, providing the structural basis for the structural and functional integration and diversity of the PSI–AcpPCI supercomplex.

### Pigment arrangement in PSI−AcpPCI

Notable differences are also found in the pigment arrangement *Symbiodinium* PSI–AcpPCI and reported red lineage PSI–LHCIs[4,13,14] (Supplementary Fig. 20a, b). PSI–AcpPCI lacks 10 Chls in PsaA (A1-A10), 10 Chls in PsaB (B2, B4-B7, B10-B14), 2 Cars in PsaA (ACar2, ACar4), and 3 Cars in PsaB (BCar1-3) (Fig. 4a, b). Instead, 8 previously unidentified Chls (Chl832-833 in PsaA, Chl818 in PsaB and Chl403-404, Chl406-408 in PsaL) and a previously unidentified Car in PsaI appears. The absence of A1-A5 and ACar1-ACar2 in PasA is attributed to the conformational changes in the C276-D294 loop of PsaA at the stromal surface (Fig. 4c, Supplementary Fig. 20c). However, 3 Chls (Chl832 in PsaA and Chl407-

408 in PsaL) are observed in the nearby positions close to the altered loop of PsaA and extended loop of PsaL. Similarly, A6-A8 are absent due to the conformational changes in the loops (A421-I431 and G213-L244) of PsaA at the lumenal surface (Fig. 4c, Supplementary Fig. 20d). A Chl (Chl833), close to A8, binds to the altered loop of PsaA. Furthermore, *Symbiodinium* PsaA loses two Chls because of the changes in the ligands of A9 and A10 from H205 and H315 in diatom PsaA to S182 and L272, respectively (Fig. 4d, Supplementary Fig. 20e).

The conformational changes in the loops of PsaB (Y261-T270 and S279-F284 at the stromal side, A215-S223 and G429-S449 at the lumenal side) also contribute to the dissociation of Chls and Cars (Fig. 4e, f). The shortening of the PsaB loop (S279-F284) and the presence of an extended N-terminal loop in AcpPCI-8 hinder the binding of B4 and B5 (Fig. 4e, Supplementary Fig. 20f). Changes in the loop region (G429-S449) result in the absence of B6 and B7 (Fig. 4e, Supplementary Fig. 20g). The absence of B4-B7, coupled with alternations in the surrounding residues, reduce hydrophobic interactions, resulting in the dissociation of B2, B11, BCar1 and BCar2 (Supplementary Fig. 20h,i). These changes also contribute to a decrease in the hydrophobic environment. The increased volume of the altered side chain and changes in their orientations further impede the binding of these pigments. *Symbiodinium*-specific substitutions in contrast to their diatom counterpart, which weaken the hydrophobic and hydrogen-bond interactions, along with the increased volume of the altered side chains, hinder the binding of B10 and B12-B14 (Fig. 4f, Supplementary Fig. 20j). Additionally, the substitution of V184 with F196 in *Symbiodinium*, in conjunction with the absence of B10, induces a

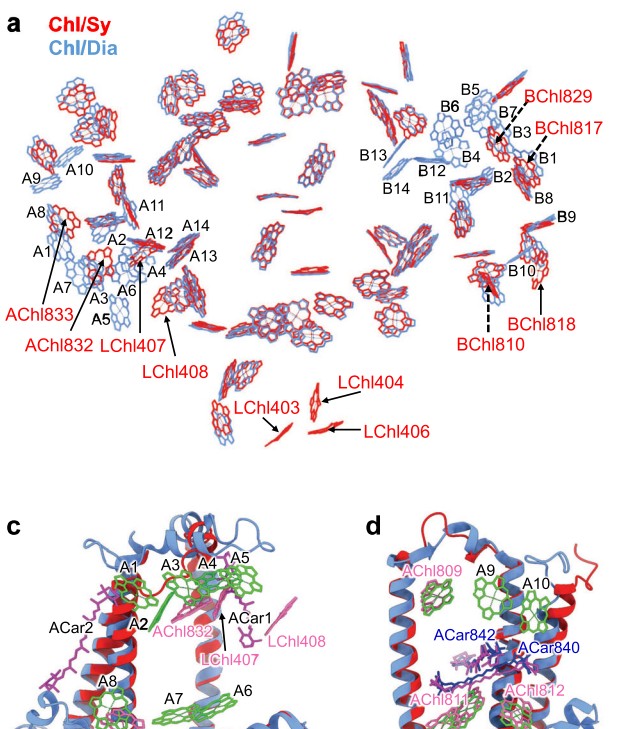

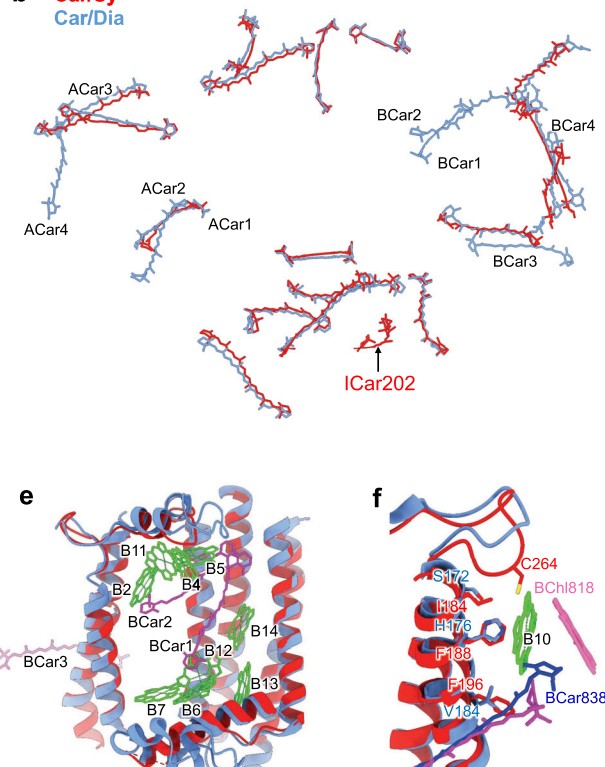

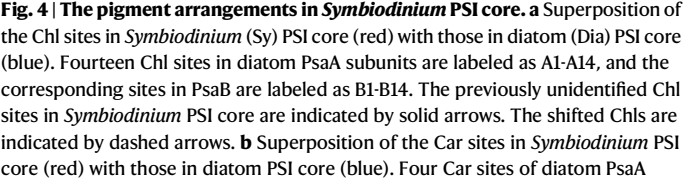

**Fig. 4 | The pigment arrangements in *Symbiodinium* PSI core. a** Superposition of the Chl sites in *Symbiodinium* (Sy) PSI core (red) with those in diatom (Dia) PSI core (blue). Fourteen Chl sites in diatom PsaA subunits are labeled as A1-A14, and the corresponding sites in PsaB are labeled as B1-B14. The previously unidentified Chl sites in *Symbiodinium* PSI core are indicated by solid arrows. The shifted Chls are indicated by dashed arrows. **b** Superposition of the Car sites in *Symbiodinium* PSI core (red) with those in diatom PSI core (blue). Four Car sites of diatom PsaA

subunits are labeled as ACar1-ACar4, and the corresponding sites in PsaB are labeled as BCar1-BCar4. The previously unidentified Car site in *Symbiodinium* PSI core is indicated by solid arrow. Superposition of the PsaA subunits (**c–d**) and PsaB subunits (**e–f**) of *Symbiodinium* (red) and diatom (blue), showing the differences in pigment associations. Chls and Cars in *Symbiodinium*, and Chls and Cars in diatom are colored in pink, blue, green, and magenta, respectively.

confirmational flipping of the head of Car838, potentially facilitating the binding of the Chl818 (Fig. 4f). The loss of BCar3 is caused by the shortening of the lumenal loop (A215-S223) of PsaB, which disturbs the hydrophobic environment (Supplementary Fig. 20k). Furthermore, changes in the surrounding residues result in the conformational shifts of Chl810, Chl817, and Chl829 (Supplementary Fig. 20l, m).

Twenty Chl-binding sites and thirteen Car-binding sites were identified in AcpPCIs, out of which twelve Chl-binding sites (301-312) and five Car-binding sites (401-405) are conserved in the reported red lineage LHCIs[2-4,13,14] (Fig. 3d, e, Supplementary Table 4). Other than the conserved sites, eight additional Chl sites (313-320) and eight additional Car sites are distributed across different AcpPCIs (Supplementary Fig. 21a–f). These specific Chl/Car-binding sites are located in the interface region between AcpPCIs and between AcpPCIs and the PSI core subunits (Supplementary Fig. 21e–f), suggesting their important role in mediating inter-subunit energy transfer. Interestingly, some Chl-binding sites, such as 301, 303, 304, 306 and 310, can accommodate either Chl $a$ or Chl $c$ (Supplementary Fig. 21b, c). Most AcpPCIs contain one or two Chl $c$, except for AcpPCI-9, which possesses five Chl $c$ molecules (Supplementary Fig. 21e). The high content of Chl $c$ in AcpPCI-9 may facilitate energy transfer from AcpPCI-9 to adjacent subunits, as Chl $c$ has higher energy than Chl $a$[3,7,8,32]. All AcpPCIs feature a potential red-shifted Chl 305/306 pair located at the same positions as other LHCIs[2-14,32]. The PSI–AcpPCI complex contains Ddx and Dtx, as confirmed by HPLC analysis (Supplementary Fig. 1e). Since Ddx and Dtx share similar structures, the putative binding sites for Dtx found at a low level were assigned as for Ddx in the PSI–AcpPCI model. The presence of Dtx indicates Ddx-to-Dtx conversion that contributes to the activation of NPQ occurred preferentially in PSI–LHCI[30], suggesting the role of PSI–LHCI in photoprotection.

Carotenoids play dual roles in the processes of excitation energy transfer: transferring excitation energy to Chls and quenching excessive excitation energy from Chls to facilitate photoprotection[33–35]. The AcpPCI-associated carotenoids have close interactions with Chls located within or around AcpPCIs (Supplementary Fig. 21k). This suggests that there is a highly efficient energy transfer between these pigment molecules. In addition, it is worth noting that *Symbiodinium* PSI–AcpPCI may possess multiple distinct energy quenching sites that are formed by the presence of previously unidentified Chls and Cars (Supplementary Fig. 21k–u). The previously unidentified Cars in PSI–AcpPCI (AcpPCI-5/Car 410, AcpPCI-8/Car 409/411/412, AcpPCI-9/Car 406/408/410, PsaI/Car 202, AcpPCI-1/Car 407/413) are in close proximity to Chls and are presumably responsible for dissipating excessive excitation energy from Chls (Supplementary Fig. 21k–q). The previously unidentified Chls (AcpPCI-9/Chl 320, AcpPCI-1/Chl 313/319, PsaB/Chl 818, PsaL/Chl 403/404/406/407/408, AcpPCI-3/Chl 317/318, AcpPCI-7/Chl 315) exhibit strong associations with Cars and have the potential to transfer excessive excitation energy to these Cars (Supplementary Fig. 21q–u). These unique structural features suggest that *Symbiodinium* PSI–AcpPCI may play a critical role in photoprotection.

## Model of energy transfer within the PSI−AcpPCI supercomplex

Based on our structural data, we calculated excitation energy transfer (EET) within the PSI−AcpPCI complex. The EET rates between all Chl pairs were calculated using the Förster theory[36]. Moreover, the generalized Förster theory, an extension of the classical Förster theory, was employed to simulate the EET rates between AcpPCIs as well as between AcpPCIs and the PSI core[37]. The EET time constant between Chls, which is the reciprocal of the EET rate, reflects the dynamics of EET processes within the PSI−AcpPCI complex.

The EET time constant map for PSI−AcpPCI reveals that the EET rates between Chls occur on the picosecond time scale (Fig. 5a), consistent with the simulated EET rates of plant PSI−LHCI[5]. Many EET rates between Chls reach sub-picosecond rates, indicating rapid energy transfer within PSI−AcpPCI. The efficiency of EET within individual AcpPCIs and within the PSI core is higher compared to that between AcpPCIs and between AcpPCIs and the PSI core (Fig. 5a). Additionally, the stromal and lumenal Chl layers exhibit efficient EET (Fig. 5b). However, EET between AcpPCI-1 and AcpPCI-2, AcpPCI-2 and AcpPCI-12, AcpPCI-7 and AcpPCI-8, and AcpPCI-9 and AcpPCI-10 is less efficient due to the large gap between them (Fig. 5a, c). Consequently, AcpPCIs can be divided into three groups for energy transfer to the PSI core: AcpPCI-2/3/4/5/6/7/12/13 (Group I), AcpPCI-8/9 (Group II), and AcpPCI-10/11/1 (Group III) (Fig. 5a, c).

In Group I, the most efficient EET occurs from AcpPCI-3/5/7 to the PSI core (Fig. 5c). Notably, the EET rate from AcpPCI-7 to the PSI core is less than 1.0 ps, suggesting a rapid energy transfer pathway. Direct energy migration from AcpPCI-4/6 to the PSI core is less efficient, but their energy can be transferred at a high rate to AcpPCI-3 and AcpPCI-7, respectively. Thus, EET from AcpPCI-4/6 to the PSI core primarily occurs through the mediation of AcpPCI-3 and AcpPCI-7 (Supplementary Table 5). AcpPCI-12 and AcpPCI-13, being farther from the PSI core, transfer energy to the PSI core via AcpPCI-3 and AcpPCI-4. Intriguingly, AcpPCI-13 utilizes three distinct pathways for energy transfer to the PSI core with similar rates of ~12 ps (Supplementary Table 5). Direct EET from AcpPCI-2 to the PSI core is inefficient, but this appears to be compensated by the AcpPCI-2–AcpPCI-3–core pathway. In Group II, AcpPCI-8 demonstrates a high-rate EET to the PSI core, while the EET from AcpPCI-9 to the PSI core is significantly slower due to the large gap between them (Fig. 5a, c). Instead, AcpPCI-9 efficiently transfers energy to the PSI core through the mediation of AcpPCI-8. In Group III, all three AcpPCIs efficiently transfer energy to the PSI core, with AcpPCI-1 (1.1 ps) and AcpPCI-10 (1.6 ps) exhibiting particularly rapid rates (Fig. 5c). Moreover, AcpPCI-11 possesses an additional efficient pathway for EET through AcpPCI-10.

It is worth noting that part of the EET map in PsaB is incomplete due to the absence of ten Chls (Figs. 4a, 5a). Energy from AcpPCI-8/9/10/11 is primarily transferred to P700 through the pathways located at the PsaI−PsaM side of PsaB. In contrast, the EET map in PsaA shows a denser arrangement despite also lacking ten Chls. The missing Chls in PsaA are situated in the marginal area and their absence is compensated by the presence of four Chls (AChl832-833, LChl407-408) (Fig. 4a). Furthermore, the interior Chls 11-14, absent in PsaB, are present in PsaA. These differences in the Chl association contribute to significant distinctions in the EET within PsaA and PsaB. Furthermore, P700 re-reduction kinetics revealed that the photochemical efficiency and electron transport rate of *Symbiodinium* PSI−AcpPCI are slightly higher than those of cryptophyte PSI−ACPI, suggesting that the reducing Chl content in the PSI core has no effect on energy transfer within *Symbiodinium* PSI (Supplementary Fig. 22).

The Chl 305/306 pair located at the stromal side is critical in mediating the EET from AcpPCI-2/6/8/9/10 to the PSI core, while Chls 311 and 312 mediate energy transfer from AcpPCI-5 and AcpPCI-4/7/11, respectively, to the PSI core at the lumenal side (Fig. 5d, Supplementary Table 6). Due to the orientation change of AcpPCI-1, its Chls 302/307/308 face the PSI core and transfer energy to the Chls 303/304/306 in PsaL. The pigments of AcpPCI-3/4 exhibit a conformational shift towards the PSI core compared to their counterparts in diatom FCPI-3/4 (Supplementary Fig. 23a, b). Among them, Chls 302-304 of AcpPCI-3 and Chls 308/312 of AcpPCI-4 form efficient EET pathways with the Chls A832 and A833, respectively. Additionally, Chl 318 of AcpPCI-3 transfers energy to Chls A824/A825 at a high rate, compensating for the loss of Chls A1, A3, and A5-A9 at the interface (Supplementary Fig. 23a, b). Chl 311 of AcpPCI-5, which is absent in FCPI-5, mediates efficient EET at the lumenal side to Chls A811/A813 of PsaA, compensating for the lack of EET pathways at the stromal side due to the absence of Chls 9/10 of PsaA (Supplementary Fig. 23c).

AcpPCI-1/3/4 exhibit changes in orientation compared to other AcpPCIs, resembling Lhcb9 in the PSI-LHCI supercomplex from moss *Physcomitrium patens*. In Lhcb9, the red Chl pair 603/609, which

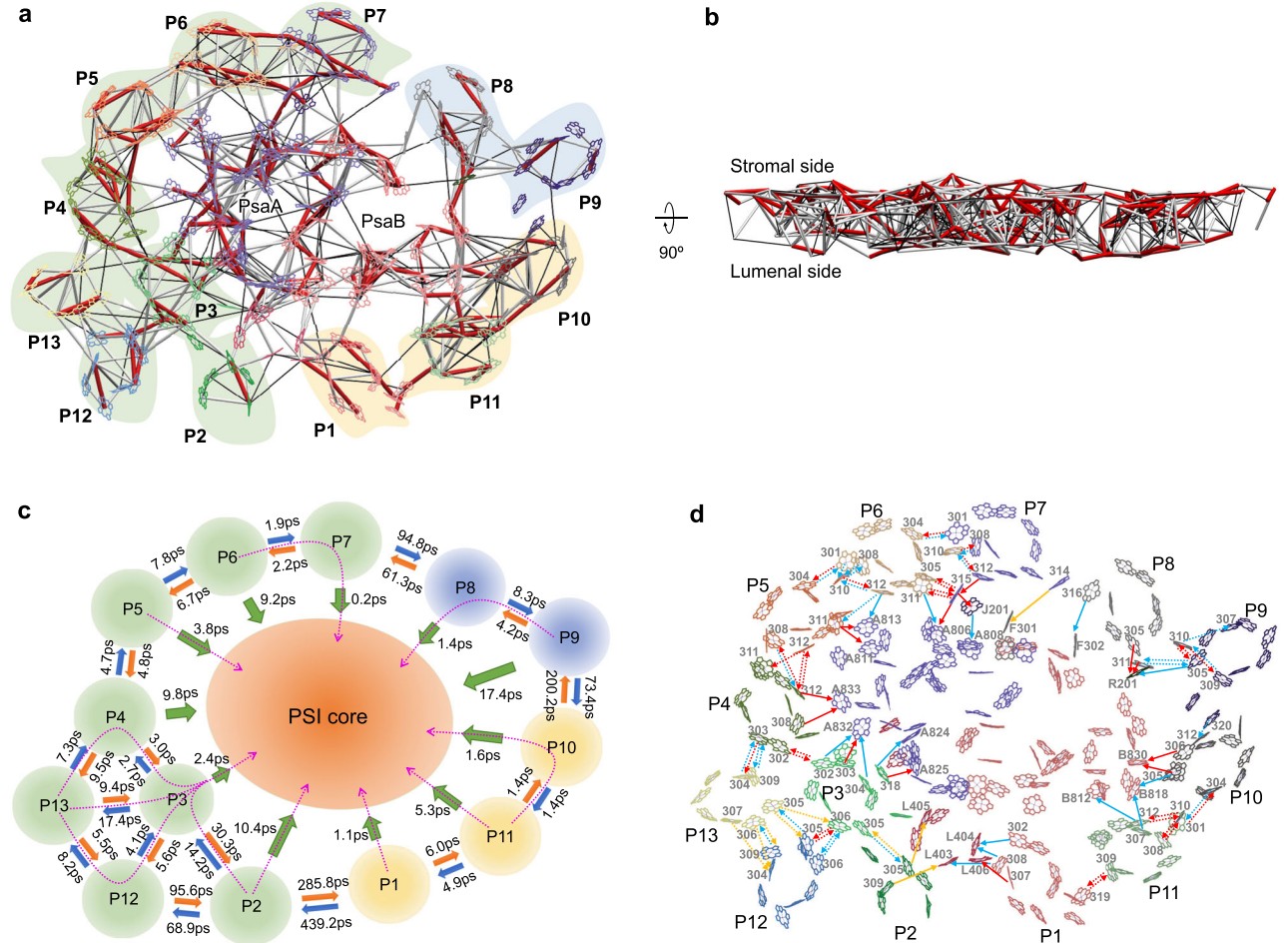

**Fig. 5 | Excitation energy transfer (EET) pathways in *Symbiodinium* PSI–AcpPCI.** Top view (**a**) and side view (**b**) of the maps of inter-pigment EET rates. The color bars correspond to rates faster than 1 ps (red), in the 1–10 ps range (thick white) and in the 10–20 ps range (thin black). Inter-pigment rates slower than 20 ps are omitted. The shadings indicate the outlines of AcpPCIs. Group I (green), Group II (blue), and Group III (orange). **c** EET rates between AcpPCIs and from AcpPCIs to PSI core. The efficient EET pathways from AcpPCIs to PSI core are indicated by magenta arrows. **d** EET between the Chls viewed from the stromal side. EET pathways from AcpPCIs to PSI core (solid arrows), between the AcpPCIs (dashed arrows). Red arrows: faster than 2 ps; blue arrows: between 2 ps and 5 ps; orange arrows: between 5 ps and 10 ps. P1-P13 represent the AcpPCI-1 to AcpPCI-13 subunits, respectively.

corresponds to the Chl pair 305/306 in AcpPCIs, is faced outward and is sandwiched by two carotenoids L2 and L3 (Supplementary Fig. 23h). This pigment arrangement was presumed to play a role in quenching excess energy in addition to light harvesting[38,39]. Intriguingly, a similar pigment arrangement was identified in AcpPCI-1, whereas the carotenoid equivalent to L3 in Lhcb9 is absent in AcpPCI-3/4 (Supplementary Fig. 23h). These findings suggest that AcpPCI-1 may function similarly to Lhcb9 in energy dissipating and light harvesting.

The previously unidentified Chl 315 in AcpPCI-7 plays a critical role in efficient EET pathways between AcpPCI-6/7 and the PSI core (Fig. 5d, Supplementary Fig. 23d). These pathways mediated by Chl 315 are absent in red algae, cryptophyte, and diatom PSI–LHCIs, indicating the efficient EET between AcpPCI-6/7 and the PSI core. The counterparts of diatom FCPI-7/Chl 405 and FCPI-8/Chl 409 are missing, reducing the lumenal energy transfer between AcpPCI-7 and 8 (Supplementary Fig. 23e). However, the Chl 314 in AcpPCI-7 at the stromal side and the Chl 316 in AcpPCI-8 at the lumenal side enhance energy transfer to the PSI core. Due to the absence of Chls B5-B7 in PsaB, energy from AcpPCI-8 is transferred to Chl R201 of PsaR through the Chl 305/306 pair and Chl 311 that is closer to Chl R201 compared to its diatom counterpart Chl 410 (Supplementary Fig. 23f). The reduced distance between Chl 311 and Chl R201 potentially promotes the energy transfer rate to compensate for the missing EET pathways resulted from the loss of Chls B5-B7. AcpPCI-8 lacks the stromal Chl 413 present in FCPI-8,

but it possesses the lumenal Chl 310 that is absent in FCPI-8, forming efficient EET pathways with the Chl 305/306 pair and Chl 309 of the shifted AcpPCI-9 (Supplementary Fig. 23f). The Chl 305/306 pair of AcpPCI-10 is closer to Chl B830 compared to those of FCPI-10, facilitating high-rate EET to Chl B830 (Supplementary Fig. 23g). The previously unidentified Chl B818 is positioned near AcpPCI-10/11 and accepts energy from the Chl 305/306 pair of AcpPCI-10 and Chl 307 of AcpPCI-11 (Supplementary Fig. 23g). Chl 312 of AcpPCI-11 which is absent in FCPI-11 transfers energy to Chl B812, thereby strengthening the EET between AcpPCI-11 and the PSI core (Supplementary Fig. 23g).

Chls 301/304/308/310/312 are the primary pigments responsible for energy transfer between AcpPCIs. However, due to the conformational shifts and rotations of AcpPCI-1/2/3/4/9/12/13 as well as the binding of previously unidentified Chls, various other Chls, such as Chls 302/303/305/306/307/309/311/315/319/320, also participate in EET between AcpPCI-1/2/3/4/7/9/12/13 and neighboring AcpPCIs (Fig. 5d, Supplementary Table 7). EET between the stromal and lumenal Chl layers of AcpPCI-4/5/6/7/8/10/11 is mainly facilitated by the Chl 305/306 pair at the stromal side and Chls 309/311/312 at the lumenal side. However, this pattern differs for AcpPCI-1/3/9/12/13, as Chls 311/312 are absent in AcpPCI-1/3/9/13 and Chl 311 exhibits a conformational shift in AcpPCI-12 (Supplementary Fig. 24). Instead, EET between the two Chl layers in AcpPCI-1 is mediated by the Chls 313/319, and in AcpPCI-3 by the Chls 317/318, the Chl 305/306 pair and Chls 303/309.

EET between the two Chl layers in AcpPCI-9/12/13 is less efficient. However, due to the rotations of these AcpPCIs, their Chls form EET networks with Chls in the opposite layers of adjacent AcpPCIs (Supplementary Fig. 24). In addition, the Chl 315 in AcpPCI-7 forms efficient EET pathways with Chl 305 in the opposite layer of AcpPCI-6.

Overall, the lack of specific Chls in the PSI core and the conservation of Chl-binding sites in AcpPCIs result in the loss of certain EET pathways that rely on those particular Chls. However, the emergence of previously unidentified Chls in the PSI core and AcpPCIs, along with the conformational rotations and shifts of AcpPCIs and Chls, leads to the formation of specific EET pathways in *Symbiodinium* PSI−AcpPCI.

Both dinoflagellates and diatoms are important primary producers in the ocean, although they have evolved distinct PSI−LHCI supercomplexes. Diatom PSI−FCPI possesses the most extensive antenna and pigment system, as well as the most complicated EET network, among all the previously reported PSI−LHCI supercomplexes. These features enable diatoms to effectively adapt to fluctuating light conditions. In contrast to diatom PSI−FCPI, *Symbiodinium* PSI−AcpPCI has reduced quantities of antennae and pigments in both the antennae and the PSI core. Nevertheless, the PSI core subunits and antennae of PSI−AcpPCI possess extended terminal domains that are absent in diatom PSI−FCPI. Furthermore, the antenna and pigment arrangements as well as the EET pathways in PSI−AcpPCI differ from those in diatom PSI−FCPI. These structural variations likely reflect the ability of dinoflagellates to adapt and survive in more stable light conditions and clearer water, as opposed to diatoms.

In summary, the cryo-EM structure of PSI−AcpPCI unraveled the architecture of the PSI−LHCI supercomplex in the symbiotic dinoflagellate *Symbiodinium* sp., which possesses specific PSI core subunits and antennae with extended terminal domains, as well as specific pigment arrangements for efficient energy transfer. These structural characteristics provide insights into the adaptive mechanisms of light harvesting and energy transfer employed by *Symbiodinium* PSI−AcpPCI, enabling them to thrive in their specialized environmental habitats.

## Methods

### Purification of the PSI−AcpPCI from *Symbiodinium*

*Symbiodinium* sp. GY-H50 (catalog number: GY-H50) was purchased from Shanghai Guangyu Biological Technology Co., Ltd. It was isolated by limiting dilution method from coral in Beibu Gulf (Guangxi province, China) and was monospecific. To verify the strain, we performed 18 S ribosomal RNA and internal transcribed spacer (ITS) gene sequencing, confirming that the strain belongs to *Symbiodinium*. The ribosomal RNA gene sequences were deposited to the National Center for Biotechnology Information database with the accession numbers PP191135 and PP191136. Primers are listed in Supplementary Table 8.

The alga was grown in 8 L of F/2 medium under continuous light at 40 $\mu$mol photon m$^{-2}$ s$^{-1}$ at 23 °C with bubbling of air. The cells at logarithmic phase were collected by centrifugation ($6000 \times g$, 10 min). The harvested cells were disrupted with glass beads in buffer A (25 mM MES-NaOH, pH 6.5, 1.0 M betaine, 10 mM MgCl$_2$). The cell debris were removed by centrifugation ($3000 \times g$, 2 min). The thylakoid membranes in supernatant were collected by centrifugation at $21,000 \times g$ for 20 min, and then washed with buffer B (25 mM MES-NaOH, pH 6.5, 1.0 M betaine, 1.0 mM EDTA). The thylakoid membranes at 0.4 mg mL$^{-1}$ Chl were solubilized with 2.8% (w/v) n-dodecyl-$\alpha$-D-maltopyranoside ($\alpha$-DDM) (Anatrace, USA) for 15 min in buffer C (25 mM MES, pH 6.5, 1.0 M betaine, 10 mM NaCl, 5.0 mM CaCl$_2$) on ice in dark with gentle stirring. The solubilized membranes were centrifuged at 21,000 $g$ for 20 min at 4 °C, and the supernatant was loaded onto a sucrose gradient of 10%-30% in buffer C containing 0.02% $\alpha$-DDM. After centrifugation at 230,500 $g$ for 20 h (Beckman SW41Ti rotor), the PSI−AcpPCI sample was collected in 20% layer, and then further purified by size-exclusion

chromatography (GE; Superose 6 Increase 10/300 GL) in buffer D (25 mM MES, pH 6.5, 0.5 M betaine, 50 mM NaCl, 5.0 mM CaCl$_2$). The sample was concentrated to 2.0 mg ml$^{-1}$ in Chl using a 100 kDa cut-off filter (Amicon Ultra; Millipore) for cryo-EM specimen preparation. All steps were performed at 4 °C under dim light.

### Characterization of PSI−AcpPCI

Absorption spectra were recorded at room temperature with a Shimadzu UV−Vis 1990 spectrophotometer[40]. To analyze the subunit composition of PSI−AcpPCI, samples were separated by 8%-16% SDS-PAGE and identified by mass spectrometry analysis (Supplementary Data 1)[13,41]. The protein bands were cut out from the gel, reduced with dithiothreitol, alkylated with iodoacetamide, and digested with trypsin, and the resulting peptide fragments were examined by liquid chromatography-tandem mass spectrometry (LC-MS/MS). The peptides were separated with Easy-nLC 1000 System equipped a phase trap column (nanoViper C18, 100 $\mu$m × 2 cm, Thermo Fisher) connected to the C18-reversed phase analytical column (75 $\mu$m × 10 cm, 3 $\mu$m resin, Thermo Fisher) and coupled to a Q Exactive mass spectrometer (Thermo Fisher). The acquired spectra were searched against the selected database using MASCOT engine (version 2.4) with the Proteome Discovery searching algorithm (version 1.4).

Pigment composition of the PSI was analyzed by LC-20AD high performance liquid chromatograph (Shimadzu, Janpan) as described previously[4]. Pigments were extracted by pre-cooled 100% acetone overnight at 4 °C protected from light. The extract was concentrated and injected into C18 reversed-phase column (Waters, Ireland). The elutes were detected by a Shimadzu photodiode array detector at 445 nm with a wavelength detection range of 300−800 nm. Seven pigments were identified based on the characteristic absorption peaks of their absorption spectra and elution profiles[42], in line with previous reports[43,44].

### Sequence analysis of the PSI−AcpPCI from Symbiodinium

Transcriptome sequencing of *Symbiodinium* sp. were performed by BioMarker (BMK) through high-throughput sequencing. Total RNA was extracted from *Symbiodinium* cells, and the cDNA Library was constructed using the NEBNext Ultra RNA Library Prep Kit for Illumina (NEB, MA, USA) following the manufacturer's recommendations. The first strand of cDNA was synthesized using the mRNA fragments as template with random hexamer primers and RNase H. DNA polymerase I and RNase H were used for synthesizing the second strand of cDNA. The synthesized double-stranded cDNA was modified by terminal repair, A-tailing, and adapter addition for hybridization. The cDNA fragments ∼ 150 bp in length were selected using the AMPure XP beads (Beckman Coulter, Beverly, USA). The cDNA was treated with USER Enzyme (NEB), and PCR was performed to obtain the final cDNA library. After clustering of the index-coded samples, the library preparations were sequenced on an Illumina HiSeq 2000 platform, and paired-end reads were generated. The transcriptome was assembled de novo based on the left.fq and right.fq using Trinity 2.5.1[45]. Sequences of the PSI core and AcpPC subunits were identified by retrieving the homologous sequences in the National Center for Biotechnology Information databases from the transcriptome sequences. The sequence alignment was done using CLC Sequence Viewer 8.0 and ESPript 3.0.

### Phylogenetic analysis

The sequences for producing phylogenetic trees were aligned with MUSCLE (default parameters), and the phylogenetic tree was constructed using MEGA X[46]. The evolutionary history was inferred using the Neighbor-Joining method[47]. The percentage of replicate trees in which the associated taxa clustered together in the bootstrap test (1000 replicates) are shown next to the branches[48]. The evolutionary distances were computed using the Poisson correction method[49] and

are in the units of the number of amino acid substitutions per site. All ambiguous positions were removed for each sequence pair (pairwise deletion option).

## P700 redox kinetics

The P700 redox kinetics of cell suspensions at a Chl concentration of $0.24 \text{ mg ml}^{-1}$ were measured with a Dual-PAM-100 (Walz) after dark adaptation. To measure the PSI photochemical efficiency, the slow kinetics of P700 were recorded. P700 was completely oxidized (Pm) by a 200 ms saturation pulse (SP) (maximum intensity at 625 nm, $10,000 \text{ μM photons m}^{-2} \text{ s}^{-1}$) after the cells were illuminated by far-red light at 720 nm ($250 \text{ μM photons m}^{-2} \text{ s}^{-1}$) for 10 s, as SP induced multiple charge separations in the PSI reaction center. SP also induced multiple charge separations in the PSII reaction center and the resulting electrons were sufficient to fully reduce the secondary electron acceptor PQ. After SP is turned off, the oxidized P700 was fully reduced (Po) with great rapidity by electrons from PSII through PQ, cytochrome (Cyt) $b_6 f$, and plastocyanin. Forty seconds after the first SP, the actinic light (AL) (maximum intensity at 625 nm, $80 \text{ μM photons m}^{-2} \text{ s}^{-1}$) was opened, and a SP was applied every 20 s. The active and open PSI reaction center was completely oxidized (Pm') by SP. The measurement was finished until the current P700 signal (P) and the Pm' were stable. The photochemical efficiency of PSI was calculated: Y(I) = (Pm'−P)/(Pm−Po). The light curve was measured using the light curve program files, showing the relationship of the electron transport rate (ETR) and photosynthetically active radiation (PAR). The step width at each radiation intensity was 30 s.

## Cryo-EM data collection and processing

Holey-carbon grids (Quantifoil Au R2/1, 200 mesh) were glow-discharged for cryo-EM sample preparation before $4.0 \text{ μL}$ of PSI-AcpPCI sample ($2.0 \text{ mg mL}^{-1}$) was applied onto the freshly prepared grids. Then the grids were blotted in Vitrobot Mark IV (Thermo Fisher Scientific) working at 100% humidity at 8 °C with a 1 s blotting time, force level of -1 and then were plunged into liquid ethane cooled by liquid nitrogen for vitrification. Data collection was performed on a 300 kV Titan Krios G3i microscope (Thermo Fisher Scientific) equipped with a K3 BioQuantum direct electron detector (Gatan Inc.) at a nominal magnification of ×81,000 corresponding to a pixel size of 0.53 Å on a super-resolution mode. 6092 movie stacks were recorded using EPU (Thermo Fisher Scientific)[50] in a defocus range of −1.2 to −2.2 μm with the energy filter slit set to 20 eV (GIF, Gatan) and a total dose of 50 $e^{-} \text{ Å}^{-2}$.

Image processing was mainly performed by cryoSPARC 3.3.1[51]. All movie stacks were patch motion corrected and binned by a factor of 2 with dose weighting. The CTF parameters for each movie were estimated by Patch CTF Estimation before automatic particle picking. After particle-extraction and two rounds of reference-free 2D classifications, 286,230 particles were selected with obvious incorrectly selected particle positions excluded from the particle set for Ab-initial reconstruction and 3D classification. A final set with 118,810 particles were subjected to homogeneous refinement. To improve the resolution of the density map, after 3D non-uniform refinement and sharpening, global (per-group) CTF refinement, local (per-particle) CTF refinement and particle subtraction followed by local refinement with soft mask for the regions of the peripheral LHCIs (distinguished by 1, 2, 3 and 4) were performed using cryoSPARC. The overall resolutions of the map were 2.8 Å and the resolution of local maps were to 2.9 Å, 3.1 Å, 3.0 Å, and 3.2 Å based on the gold-standard FSC 0.143.

## Model building and refinement

To build the model of *Symbiodinium* PSI−AcpPCI supercomplex, the PSI core and FCPIs in the innermost layer of diatom PSI−FCPI (PDB ID: 6LY5)[4] were rigid body fitted into the 2.80 Å overall cryo-EM map and locally refined map using UCSF Chimera[52]. The amino acid residues of each chain were subsequently corrected by referring to the sequence of the counterpart in *Symbiodinium* from transcriptome sequencing. The previously unidentified subunit PsaT and PsaU and the additional terminal domains of the core subunits and AcpPCIs were constructed by de novo model building with COOT[53]. The potential sequences of AcpPCIs were searched from the transcriptome by blast using their homologous sequences from diatom (FCPIs). The potential sequences of PsaT and PsaU were searched from the transcriptome by blast using the estimated sequences of PsaT and PsaU based on their cryo-EM density maps (Supplementary Fig. 3). As the side chains of residues could not be identified precisely based on the cryo-EM density map at the current resolution, several sequences of PsaT and PsaU were estimated and each sequence is about 20 residues. Then, each of the potential sequences of PsaT, PsaU, and AcpPCIs was manually fitted into the density map, and amino acids with distinguishable side chains (such as phenylalanine, tyrosine, tryptophan, arginine, and glycine) were used to identify the proper sequence from the potential sequences (Supplementary Fig. 3). Chl *a* and Chl *c* were distinguished by the density map corresponding to the phytol chain for Chl *a*, and the planarity of C-18[1], C-18, C-17 and C-17[1] resulting from the C-18 = C-17 double bound for Chl $c$[3,13,54]. Given the current resolution of the density map, it's impossible to differentiate between diadinoxanthin and diatoxanthin. Therefore, the binding sites for both the putative diadinoxanthin and putative diatoxanthin were assigned as diadinoxanthin. The geometrical restraints of pigments were generated from the Grade Web Server, and all the residues and cofactors were manually adjusted with COOT. The constructed model was refined against the cryo-EM map by Phenix real-space refinement[55]. After real-space refinement, manual correction and adjustment was performed with COOT. Manual correction and Phenix refinement were carried out iteratively to improve the model quality. The geometries of the structural model were evaluated with Phenix and the statistics were summarized in Supplementary Table 1.

## Simulation of the excitation energy transfer in PSI−AcpPCI

Excitation energy transfer within PSI−AcpPCI was studied in the limit of the Förster theory[36]. The energy transfer rate T (in $\text{ps}^{-1}$) values were computed using the following equation,

$$T_{mn} = \frac{2\pi}{\hbar} \left| V_{mn} \right|^2 \int_{-\infty}^{\infty} d\omega F_m(\omega) A_n(\omega) \tag{1}$$

The electron coupling strength $V_{mn}$ between pigments was calculated using the TrEsp method, which incorporates environmental screening factors[56,57].

$$V_{mn} = f \cdot \sum_{l \in m, l' \in n}^{L} \frac{q_l^T \cdot q_{l'}^T}{\left| R_{l,l'} \right|} \tag{2}$$

$$f = \begin{cases} 1; R \leq 6.6 \\ A \cdot \exp(-\beta R) + 0.54; 6.6 < R < 20 \\ 0.54; 20 \leq R \end{cases} \tag{3}$$

Here, $l$ refers to the $l$-th atom of pigment molecule m, $q^T$ denotes the transition charge of that atom, and $R_{l,l'}$ represents the distance between the corresponding atoms in the pigment molecule. $f$ is environmental screening factor.

In addition, the term $F_m(\omega)$ refers to the emission spectrum of chromophore site m, while $A_n(\omega)$ represents the absorption spectrum of chromophore site n. A Gaussian broadening function was employed to simulate the spectral properties of pigments, as demonstrated

below[58]:

$$F_m(\omega) = \frac{1}{\sigma_m\sqrt{2\pi}} \exp\left\{-\frac{(\omega_m - S - \omega)^2}{2\sigma_m^2}\right\} \quad (4)$$

$$A_n(\omega) = \frac{1}{\sigma_n\sqrt{2\pi}} \exp\left\{-\frac{(\omega_n - \omega)^2}{2\sigma_n^2}\right\} \quad (5)$$

S is the Stokes shift, and $\sigma$ represents the full width at half maximum (FWHM).

Furthermore, the generalized Förster theory, an extension of the classical Förster theory, was employed to simulate the energy transfer rates between LHCs as well as between LHC and PSI core pigment aggregates[37]:

$$k_{DA} = \frac{2\pi}{\hbar} \sum_{\alpha \in D} \sum_{\beta \in A} \frac{\exp\left(-\frac{\varepsilon_\alpha^D}{k_B T}\right)}{Z} \left|V_{\alpha\beta}^{DA}\right|^2 \int d\omega S_\alpha^D(\omega) S_\beta^A(\omega) \quad (6)$$

D refers to the donor aggregate, while A represents the acceptor aggregate. $\alpha$ and $\beta$ denote the exciton states of the donor or acceptor aggregate, respectively. Specifically, $\alpha$ refers to the exciton state index of the donor aggregate, while $\beta$ refers to the exciton state index of the acceptor aggregate. $\varepsilon$ denotes the eigenvalue of the Hamiltonian for the donor or acceptor aggregate. $Z$ is the partition function, given by $Z = \sum_\alpha \exp(-\frac{\varepsilon_\alpha^D}{k_B T})$. $V_{\alpha\beta}^{DA}$ represents the electronic coupling between excitonic states of the donor and acceptor aggregates.

Gaussian16 software (Gaussian, Inc. Wallingford, CT, USA) and custom Python scripts [https://doi.org/10.5281/zenodo.10791187][59] were used to calculate the above parameters. CAM-B3LYP/6-31 G* was used and the keyword Iop (9/40 = 4) was added to the calculation to print out more detailed configuration coefficients for subsequent calculation of electronic coupling. When simulating the spectrum, the S and FWHM of Chl $a$ were 160 cm$^{-1}$ and 240 cm$^{-1}$[60]. In addition, the spectral overlap between Chl $c$ and Chl $a$ was replaced with experimental data.

### Reporting summary

Further information on research design is available in the Nature Portfolio Reporting Summary linked to this article.

## Data availability

The cryo-EM map and atomic coordinates generated in this study have been deposited in the Protein Data Bank and the Electron Microscopy Data Bank under the accession numbers of 8JJR and EMD-36366, respectively. The atomic coordinates data used in this study are available in the Protein Data Bank database under accession code 6LY5, 7Y5E, and 7Y7B. The 18 S ribosomal RNA and internal transcribed spacer (ITS) gene sequences of *Symbiodinium* sp. GY-H50 have been deposited to the National Center for Biotechnology Information database with the accession numbers PP191135 and PP191136. The ribosomal RNA gene sequences of PsaT and PsaU have been deposited to the National Center for Biotechnology Information database with the accession numbers PP196340 and PP196339, respectively. Source data for Supplementary Fig. 1b, 1c, 1e, and Supplementary Fig. 22a, 22b are provided in the Source Data file. Source data are provided with this paper.

## Code availability

The custom Python scripts generated in this study have been deposited in GitHub [https://doi.org/10.5281/zenodo.10791187][59].

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

## Acknowledgements

We are grateful to Dian-Li Zhao, Ceng Gao and Xiao-Wei Li (Cryo-EM facility for Marine Biology, Laoshan laboratory) for cryo-EM data collection. We thank Dr. Fang Zhao for assistance in data analysis and Xiang-Mei Ren for assistance in HPLC (State Key Laboratory of Microbial Technology, Shandong University, Qingdao, China). We thank Xiao-Ju Li (State Key Laboratory of Microbial Technology, Shandong University, China) and Mei-Ling Sun (College of Marine Life Sciences, Ocean University of China) for their assistance in TEM. Numerical computations were performed on Hefei advanced computing center. This work was supported by National Key R&D Program of China (2023YFA0914600 to L.-S.Z. and L.-N.L., 2021YFA0909600 to L.-N.L. and C.-Y.L., 2022YFC2807500 to C.-Y.L.), National Natural Science Foundation of China (32100200 to L.-S.Z., 32070109 to L.N.L., 91851205 to Y.-Z.Z.), Program of Shandong for Taishan Scholars (tspd20181203 to Y.-Z.Z.), Natural Science Foundation of Shandong Province, China (ZR2020QC024, L.-S.Z.), Royal Society (URF\R\180030 to L.-N.L.), and Biotechnology and Biological Sciences Research Council (BBSRC) (BB/V009729/1 and BB/R003890/1 to L.-N.L.).

## Author contributions

L.-S.Z., Y.-Z.Z. and L.-N.L. conceived the project; N.W., L.-S.Z., F.-Y.H. and G.-M.L. performed the sample preparation and characterization; K.L. and L.-S.Z. collected the cryo-EM data. K.L. processed the cryo-EM data and reconstructed the cryo-EM density map. N.W., L.-S.Z. and C.-Y.L. built the structure model and refined the structure. J.-P.G. and J.G. performed computational simulations of EET. N.W., L.-S.Z., J.G., L.-N.L., X.-L.C., J.-P.G. and C.-Y.L. analysed the data. L.-S.Z., N.W., X.-L.C., Y.-Z.Z., and L.-N.L. wrote the manuscript paper with contributions from all other authors.

## Competing interests

The authors declare no competing interests.
