## [Peer Review File · Nature Communications]

Architecture of symbiotic dinoflagellate photosystem I–light
-harvesting supercomplex in *Symbiodinium*REVIEWER COMMENTS

Reviewer #1 (Remarks to the Author):

Zhao et al. determine the single-particle cryo-EM structure of the photosystem I-PCPI supercomplex that is found in a *Synbiodinium* species, which is a dinoflagellate that forms an endosymbiosis with corals. The structure is compared primarily to similar PSI supercomplexes from red algae. The authors identify a number of interesting characteristics. First, the authors identify two new unexpected subunits bound to the complex based on identifying their structure in the cryo-EM map. Second, the authors identify that some looping regions provide novel interactions due to them being much larger than those found in red algae. Third, various chlorophyll sites are different between the dinoflagellate and red algal structures, which substantially influences the energy transfer pathways. A highlight of the manuscript is the calculation and discussion of how energy is transferred through the chlorophyll sites, and how it is different than red algae. This especially sets the manuscript apart compared to other manuscript that report only structural features and speculate on energy transfer based solely on chromophore distances. I also note that the manuscript is very well written. I am especially impressed with the introduction that nicely links the structural importance to the ecological context. The manuscript is thorough, timely, and the conclusions drawn are substantiated. Thus, I enthusiastically recommend the manuscript for publication pending some outstanding issues listed below.

I ask that the authors provide more details on how the sequences of PsaT and PsaU were identified. I think that at least a sentence or two should be added to the results section to clarify this. To my understanding, the authors identified certain amino acids based on the cryo-EM map, and then did a blastp to search for likely sequences. If not, please clarify. If so, please state which residues, and list some similar sequences that are also candidates. It would be even more helpful if the authors showed a supplementary figure showing the confidently assigned residues.

Line 54: typo – ... from a primordial endosymbiont.

Line 469: I suggest replacing the word “junk” with “incorrectly selected particle positions” or something like that. The word junk is probably not accessible to a broad audience.

Line 469: Some numbers greater than 999 have commas (e.g., 118,810) and some do not (e.g., 286230). I suggest adding commas to numbers above 999.

Line 536: The PDB code and the EMD code need to be switched so that they correspond correctly to the first part of the sentence.

Line 716: In Figure 5, please add what the green shading means in panel a. Do they correspond to the three groups I/II/III? If so, please change line 284 to 5a instead of 5c. The authors could also add labels for each of the groups in 5a that would make the figure more accessible to the reader.

Reviewer #2 (Remarks to the Author):

The manuscript reports the characterization of the photosystem I complex (core + light-harvesting complex) in a coral symbiont alga (Symbiodiniaceae) using the CryoEM technology. The composition and structural arrangement of the constituent subunits are dissected using this powerful technique. The speed of electron transfer between the components in the photosystem complex was also measured using rapid light flash and fluorescence technology. As a non-CryoEM-expert, I am not entirely qualified to evaluate how accurate the identification and organization modeling of the subunits are and how well-supported the conclusions are. I marvel at the structural details of the photosystem that are revealed, however. I am more confident in commenting on the biological implications of the findings.

It is very nice to know that the complex shares similarities with its counterpart in diatoms. The absence and presence of subunits such as PsaK, PsaO, and PsaR across different lineages of algae are of interest. The finding of PsaT and PsaU, which were previously unidentified, and that they are also shared with diatoms and some other red-family algae is particularly interesting. However, how the sequences of these two subunits are determined is not so clear. Given that there is no prior knowledge (I assume that they were not unknown in any photosynthetic organisms), how did the CryoEM information lead to the identification of the proteins from the large pool of proteins that are in these two subunits? Was it screened by “try and error” one by one?

My more significant reservation is with the limited functional insights that the findings seem to provide. Which biological question or which part of the unique biology of dinoflagellates or the specific species of Symbiodiniaceae can be explained by the findings? What advances in dinoflagellate biology or ecology do these findings bring? Can they explain why this species can form a symbiotic relationship with corals?

I also have a few specific comments, which are given below.

Lines 95-98: there is no data for non-symbiotic Symbiodiniaceae or other dinoflagellates, this statement is not so appropriate. The same issue can be found elsewhere in the manuscript when the findings are characterized as “unique”.

Supplementary Table 3 needs sources of information.

Miss Chls: are those Chls lost due to sample processing or do they represent a physiological state?

What are the “new” Chls? What is the “newly identified Chl xxx”?

Reviewer #3 (Remarks to the Author):

Symbiodinium, belonging to the dinoflagellates, plays a crucial role in the symbiotic relationship with coral, vitalizing reefs. However, environmental stress often triggers widespread coral bleaching events, severely impacting these biodiverse ecosystems. It has been reported that the Symbiodinium PSI-LHCI

supercomplex may be involved in the photoprotection process. In this manuscript, the authors describe the cryo-electron microscopy structure of PSI-PCPI from *Symbiodinium* and exhibit distinctive structural features compared to their red lineage counterparts, including the novel subunits, sequence differences, conformational changes and pigment distributions. The results provide a solid structural basis for unraveling the mechanisms of efficient energy transfer pathways in PSI-PCPI supercomplexes and new insights into the evolutionary diversity of PSI-LHC supercomplexes among different photosynthetic organisms. Overall, this is a very nice piece of work that advances the field. I have some concerns that the authors should address before publication.

1. About the PsaU subunit:

1) Is the PsaU exclusive in dinoflagellate?

2) Are the LPS1 in red algal PSI-LHC and PsaU in PSI-PCPI belong to homologs since they share a similar occupation on the luminal side of the PSI core?

2. Line 137: The PsaI C-terminal extension may also cause a shortened N-terminus of PsaB.

1) The author should also cite the Supplementary Fig. 7 to show the shortened N-terminus of PsaB.

2) The shortened N-terminus should not be caused by the C-terminus of PsaI, and the word 'cause' could be substituted by 'adapt to' or something else.

3. Line 146 and Supplementary Fig. 12g: "The luminal loops of PsaB (A215-S223 and G429-S449) are shortened, mirroring their 147 counterparts in PsaA (G213-L244 and A421-I431). This prevents interference with the extended N148 terminus of PsaF and the luminal loops of PsaR and PCPI-10 (Supplementary Fig. 12g)". The reductions of the loops are not likely to belong to *Symbiodinium* PsaB according to Supplementary Fig. 12g, while this sentence seems to assign the loops to *Symbiodinium* PsaB. It is confused, the author needs to make some corrections and clarifications.

4. Line 142-143: 'The extended N-terminus interacts with the N-terminus of PsaI...' should be modified to 'The extended N-terminus of PsaM interacts with the N-terminus of PsaI...'

5. About PsaT (Lane 152-153): 'PsaS subunit found in diatom PSI cores occupies a similar position to PsaT.'

1) How about the sequence similarity between PsaS and PsaT?

2) Are there any regularities in the interaction patterns between PsaS/PsaT with the PSI core?

6. Lane 234, '...loop region (G429-S449) result in the absence of B6 and B7 (Fig. 5e ...' should be modified to Fig. 4e

7. About PCPI-1 and PCPI-4:

It is reported that the Lhcb9 in *Physcomitrium patens* PSI-LHC supercomplexes adopts an orientation opposite to the other LHCl monomers, its red Chl pair is faced outward and sandwiched by two carotenoids, the pigments arrangement may play a role in quenching excess energy in addition to light harvesting (Sun, H et. al, Nature plants, 2023). It seems similar to the PCPI-1 and PCPI-4, which rotated by 180 and 90 degrees, respectively. Does this pigment arrangement also exist in PCPI-1 or PCPI-4?

8. About the material: Does the structure of the complexes change when algae transition from a free-

living to a symbiotic lifestyle?

9. Supplementary Table. 3/ PsaR also exist in red algal PSI-LHC (You et.al, Nature, 2023), the author should revise it in this table.

Responses to Reviewers' Comments

Reviewer #1:

Zhao et al. determine the single-particle cryo-EM structure of the photosystem I-PCPI supercomplex that is found in a Symbiodinium species, which is a dinoflagellate that forms an endosymbiosis with corals. The structure is compared primarily to similar PSI supercomplexes from red algae. The authors identify a number of interesting characteristics. First, the authors identify two new unexpected subunits bound to the complex based on identifying their structure in the cryo-EM map. Second, the authors identify that some looping regions provide novel interactions due to them being much larger than those found in red algae. Third, various chlorophyll sites are different between the dinoflagellate and red algal structures, which substantially influences the energy transfer pathways. A highlight of the manuscript is the calculation and discussion of how energy is transferred through the chlorophyll sites, and how it is different than red algae. This especially sets the manuscript apart compared to other manuscript that report only structural features and speculate on energy transfer based solely on chromophore distances. I also note that the manuscript is very well written. I am especially impressed with the introduction that nicely links the structural importance to the ecological context. The manuscript is thorough, timely, and the conclusions drawn are substantiated. Thus, I enthusiastically recommend the manuscript for publication pending some outstanding issues listed below.

Reply: We sincerely appreciate Reviewer 1's extremely positive comments on our work and manuscript.

1. I ask that the authors provide more details on how the sequences of PsaT and PsaU were identified. I think that at least a sentence or two should be added to the results section to clarify this. To my understanding, the authors identified certain amino acids based on the cryo-EM map, and then did a blastp to search for likely sequences. If not, please clarify. If so, please state which residues, and list some similar sequences that are also candidates. It would be even more helpful if the authors showed a supplementary figure showing the confidently assigned residues.

Reply: Thanks for raising this important point and excellent suggestions. We have added additional panels (Supplementary Fig. 3b-3g) to clarify how we identified sequences of PsaT and PsaU. First, we selected small regions of the density maps of PsaT and PsaU with higher resolution (Supplementary Fig. 3b, 3e), and amino acid sequences were manually speculated based on the selected map (Speculation sequence) (Supplementary Fig. 3c, 3f). Then, blastp against the transcriptome sequences was performed using the speculation sequences to identify potential sequences (Blastp sequence) (Supplementary Fig. 3c, 3f). Each of the potential sequences was manually fitted into the selected map. Amino acids with distinguishable side chains, such as phenylalanine, tyrosine, tryptophan, arginine, and glycine, were used to identify the proper sequence from the candidates (Supplementary Fig. 3b, 3e). The sequences were further confirmed by the suitability of the side chains for the density map in other regions (Supplementary Fig. 3d, 3g). We have added relevant descriptions in RESULTS AND DISCUSSION (lines 90-99) of the revised manuscript.

2. Line 54: typo – ... from a primordial endosymbiont.

Reply: Thanks for pointing this out. We have corrected the mistake.

3. Line 469: I suggest replacing the word “junk” with “incorrectly selected particle positions” or something like that. The word junk is probably not accessible to a broad audience.

Reply: We have revised the sentence as suggested by the reviewer.

4. Line 469: Some numbers greater than 999 have commas (e.g., 118,810) and some do not (e.g., 286230). I suggest adding commas to numbers above 999.

Reply: We have added commas to numbers above 999.

5. Line 536: The PDB code and the EMD code need to be switched so that they correspond correctly to the first part of the sentence.

Reply: We have switched the PDB code and the EMD code.

6. Line 716: In Figure 5, please add what the green shading means in panel a. Do they correspond to the three groups I/II/III? If so, please change line 284 to 5a instead of 5c. The authors could also add labels for each of the groups in 5a that would make the figure more accessible to the reader.

Reply: Thanks for the suggestions. The shading in Fig. 5a represents the outlines of PCPIs. We have now recolored Fig. 5a and 5c to indicate the Group I (green), Group II (blue), and Group III (orange) of PCPIs. We have also added the color codes in figure legend and referred Fig. 5a in the corresponding sentence (lines 294-295).

Reviewer #2:

The manuscript reports the characterization of the photosystem I complex (core + light-harvesting complex) in a coral symbiont alga (Symbiodiniaceae) using the CryoEM technology. The composition and structural arrangement of the constituent subunits are dissected using this powerful technique. The speed of electron transfer between the components in the photosystem complex was also measured using rapid light flash and fluorescence technology. As a non-CryoEM-expert, I am not entirely qualified to evaluate how accurate the identification and organization modeling of the subunits are and how well-supported the conclusions are. I marvel at the structural details of the photosystem that are revealed, however. I am more confident in commenting on the biological implications of the findings.

It is very nice to know that the complex shares similarities with its counterpart in diatoms. The absence and presence of subunits such as PsaK, PsaO, and PsaR across different lineages of algae are of interest. The finding of PsaT and PsaU, which were previously unidentified, and that they are also shared with diatoms and some other red-family algae is particularly interesting.

Reply: We sincerely appreciate Reviewer 2's positive comments on our work.

1. How the sequences of these two subunits are determined is not so clear. Given that there is no prior knowledge (I assume that they were not unknown in any photosynthetic organisms), how did the CryoEM information lead to the identification of the proteins from the large pool of proteins that are in these two subunits? Was it screened by "try and error" one by one?

Reply: Please see our responses to the same question of Reviewer 1 above. We have added additional panels (Supplementary Fig 3b-3g) to clarify the determination of the sequences of PsaT and PsaU, and have added relevant descriptions in RESULTS AND DISCUSSION of the revised manuscript.

2. My more significant reservation is with the limited functional insights that the findings seem to provide. Which biological question or which part of the unique biology of dinoflagellates or the specific species of Symbiodiniaceae can be explained by the findings? What advances in dinoflagellate biology or ecology do these findings bring? Can they explain why this species can form a symbiotic relationship with corals?

Reply: Dinoflagellates are a prominent clade within the red lineage, and have ecological and evolutionary significance because they thrive as primary producers in diverse aquatic habitats and the plastids of photosynthetic dinoflagellates originated from a primordial endosymbiont. However, despite studies on red algae, cryptophytes, and diatoms, how *Symbiodinium* orchestrates the evolution of PSI-LHCI to drive efficient photosynthesis remains poorly understood.

Our structural analysis, combined with computational calculation and spectroscopic analysis, reveals the structural basis for distinctive protein organization, pigment arrangement, and excitation energy transfer (EET) pathways of *Symbiodinium* PSI-PCPI, which enable efficient light harvesting and energy transfer both between antennas and from peripheral antennas to the PSI core. P700 re-reduction kinetics revealed that *Symbiodinium* PSI-PCPI possesses high photochemical efficiency and electron transport rate. Moreover, to protect the photosynthetic apparatus from excess light in the shallow ocean, excitation energy can be transferred directly from PSII to PSI (termed "spillover") or dissipated as thermal energy (termed nonphotochemical quenching, NPQ) at P700⁺ in *Symbiodinium* (Gorbunov *et al.*, *Limnol Oceanogr*, 2001, 46: 75-85; Slavov *et al.*, *Biochim Biophys Acta*, 2016, 1857: 840-847). Our structural data demonstrate a large amount of diadinoxanthin (Ddx) and diatoxanthin (Dtx) in *Symbiodinium* PSI-PCPI, and the conversion of Ddx to Dtx contributes to the activation of NPQ (Kato *et al.*, *Plant Physiol*, 2020, 183: 1725-1734). In addition, the new Chl/Car-binding sites are located in the interface region between different PCPIs and between PCPIs and the PSI core, and form close interactions with surrounding pigments. These findings, together with the specific pigment organisation found in PCPI-1 (see our responses to Reviewer 3 below), indicate the important roles of PSI-PCPI in light harvesting and photoprotection. Our study elucidates the structural basis for the efficient energy transfer and quenching in PSI-PCPI, which is critical for

Symbiodinium to adapt to high light in the shallow ocean and to satisfy the nutrition needs of their coral hosts, stabilizing the trophic foundation for symbiotic ecosystem in coral reefs.

We agree with both Reviewer 1 and Reviewer 3 that our study provides a solid structural basis for unravelling the mechanisms of efficient energy transfer pathways and energy quenching in the PSI-PCPI supercomplex from *Symbiodinium* and provides new insights into the evolutionary diversity of PSI-LHC supercomplexes among different photosynthetic organisms. We also agree that extensive spectroscopic analysis of *Symbiodinium* PSI-PCPI remains to be investigated in our future studies to provide functional insight into the special PSI-LHCI complex.

3. Lines 95-98: there is no data for non-symbiotic Symbiodiniaceae or other dinoflagellates, this statement is not so appropriate. The same issue can be found elsewhere in the manuscript when the findings are characterized as “unique”.

Reply: Thanks for the suggestion. We have replaced “unique” with “specific” or “novel” in the revised manuscript. Despite the potential structural variations in species, our structural data do reveal remarkably distinctive protein organization, pigment arrangement, and EET pathways of *Symbiodinium* PSI-LHCI, relative to its red lineage counterparts (red algal PSI-LHCR, cryptophyte PSI-ACPI, and diatom PSI-FCPI). We agree that the structural details of PSI-LHCIs of non-symbiotic Symbiodiniaceae or other dinoflagellates remain to be characterised in future studies.

4. Supplementary Table 3 needs sources of information.

Reply: Thanks for the suggestion. We have cited relevant references in Supplementary Table 3.

5. Miss Chls: are those Chls lost due to sample processing or do they represent a physiological state?

Reply: We believe that the loss of Chls represents the physiological state of *Symbiodinium* PSI-LHCI. First, our sample processing was performed at 4 °C under dim light, and mild non-ionic detergent n-dodecyl- α -D-maltopyranoside (α -DDM) was used for membrane solubilization to reduce the disturbance of the sample. Second, the absence of Chls in the *Symbiodinium* PSI core is ascribed to the conformational alterations of apoproteins (Fig. 4, Supplementary Fig. 20). When compared to PSI-LHCI of other photoautotrophs, we determined distinct structural changes of apoproteins in *Symbiodinium* PSI-PCPI, including variations in loop lengths, conformational changes in the loop structures, and alternations in the residues surrounding Chls. These alterations lead to insufficient space, lack of ligands, and reduction in hydrophobic and hydrogen-bond interactions for Chl binding. In summary, the loss of numerous Chls results in changes in the pigment structure, which may contribute to the physiological adaptation to specific symbiotic environments.

6. What are the “new” Chls? What is the “newly identified Chl xxx”?

Reply: Both “new” Chls and “newly identified Chl xxx” indicate the Chls that are unique to this *Symbiodinium* PSI-PCPI which are absent in other PSI-LHCIs. We have replaced “newly identified” with “new” to avoid confusion.

Reviewer #3:

Symbiodinium, belonging to the dinoflagellates, plays a crucial role in the symbiotic relationship with coral, vitalizing reefs. However, environmental stress often triggers widespread coral bleaching events, severely impacting these biodiverse ecosystems. It has been reported that the *Symbiodinium* PSI-LHCI supercomplex may be involved in the photoprotection process. In this manuscript, the authors describe the cryo-electron microscopy structure of PSI-PCPI from *Symbiodinium* and exhibit distinctive structural features compared to their red lineage counterparts, including the novel subunits, sequence differences, conformational changes and pigment distributions. The results provide a solid structural basis for unraveling the mechanisms of efficient energy transfer pathways in PSI-PCPI supercomplexes and new insights into the evolutionary diversity of PSI-LHC supercomplexes among different photosynthetic organisms. Overall, this is a very nice piece of work that advances the field. I have some concerns that the authors should address before publication.

Reply: We appreciate Reviewer 3’s highly positive comments on our work.

1. About the PsaU subunit:

1) Is the PsaU exclusive in dinoflagellate?

Reply: We obtained homologous sequences of PsaU from the National Center for Biotechnology Information databases. Our sequence alignment analysis showed that PsaU is exclusive in marine dinoflagellates belonging to the genus *Symbiodinium*, with a high degree of conservation (Supplementary Fig. 9i). This was stated in Page 5, line 124-125.

2) Are the LPS1 in red algal PSI-LHC and PsaU in PSI-PCPI belong to homologs since they share a similar occupation on the lumenal side of the PSI core?

Reply: Thanks for the comment. Unfortunately, the amino acid sequence of L_{PS1} in red algal PSI-LHC has not been determined. Therefore, we were unable to perform the sequence comparison of L_{PS1} and PsaU. However, the length and structure of L_{PS1} are remarkably distinct from PsaU, although they share similar binding positions within PSI-LHCI (Supplementary Fig. 8). We will analyse the sequence similarity of L_{PS1} and PsaU in future once the sequence of L_{PS1} is identified.

2. Line 137: The PsaI C-terminal extension may also cause a shortened N-terminus of PsaB.

1) The author should also cite the Supplementary Fig. 7 to show the shortened N-terminus of PsaB.

Reply: Thanks for the suggestion. As suggested by the reviewer, we have referred Supplementary Fig. 7 in the corresponding sentence (lines 147-148) of the revised manuscript.

2) The shortened N-terminus should not be caused by the C-terminus of PsaI, and the word ‘cause’ could be substituted by ‘adapt to’ or something else.

Reply: Thanks for pointing this out. We have replaced “cause” with “adapt to” as suggested.

3. Line 146 and Supplementary Fig. 12g: “The lumenal loops of PsaB (A215-S223 and G429-S449) are shortened, mirroring their counterparts in PsaA (G213-L244 and A421-I431). This prevents interference with the extended N terminus of PsaF and the lumenal loops of PsaR and PCPI-10 (Supplementary Fig. 12g)”. The reductions of the loops are not likely to belong to *Symbiodinium* PsaB according to Supplementary Fig. 12g, while this sentence seems to assign the loops to *Symbiodinium* PsaB. It is confused, the author needs to make some corrections and clarifications.

Reply: As shown in Supplementary Fig. 12g, the PsaB subunits of *Symbiodinium* (red), red algae (gray), cryptophyte (orange), and diatom (blue) are superimposed to illustrate the contraction in length and the conformational changes of the lumenal loops of *Symbiodinium* PsaB (A215-S223 and G429-S449). We have revised Supplementary Fig. 12g by adding labels and highlighting the loop regions with dashed oval for better comprehension.

4. Line 142-143: ‘The extended N-terminus interacts with the N-terminus of PsaI....’ should be modified to ‘The extended N-terminus of PsaM interacts with the N-terminus of PsaI...’

Reply: We have clarified this sentence as suggested by the reviewer.

5. About PsaT (Lane 152-153): ‘PsaS subunit found in diatom PSI cores occupies a similar position to PsaT.’

1) How about the sequence similarity between PsaS and PsaT?

Reply: The sequence comparison of PsaS and PsaT can not be performed unfortunately, due to the fact that the amino acid sequence of PsaS has not been determined in the diatom PSI-FCPI structure given its peripheral location (Xu et al., *Nat Commun* 2020, 11: 5081). Nevertheless, as shown in Fig. 2f and stated in our manuscript, the structures of PsaS and PsaT show a low similarity. We will analyse the sequence similarity of PsaS and PsaT in future once the sequence of PsaS is determined.

2) Are there any regularities in the interaction patterns between PsaS/PsaT with the PSI core?

Reply: Although PsaS and PsaT locate at similar positions on the PSI core and have close interactions with PsaB/C/D/L, their detailed binding sites with PsaB/C/D/L are distinct given their low structural similarity (Fig. 2f). Moreover, PsaT also forms interactions with the extended C-terminus of *Symbiodinium* PsaI, which is absent for PsaS in diatom PSI.

6. Line 234, ‘...loop region (G429-S449) result in the absence of B6 and B7 (Fig. 5e ...’ should be modified to Fig. 4e

Reply: Thanks for pointing this out. We have corrected the typo in the revised manuscript.

7. About PCPI-1 and PCPI-4: It is reported that the Lhcb9 in *Physcomitrium patens* PSI-LHC supercomplexes adopts an orientation opposite to the other LHCI monomers, its red Chl pair is faced outward and sandwiched by two carotenoids, the pigments arrangement may play a role in quenching excess energy in addition to light harvesting (Sun, H et. al, Nature plants, 2023). It seems similar to the PCPI-1 and PCPI-4, which are rotated by 180 and 90 degrees, respectively. Does this pigment arrangement also exist in PCPI-1 or PCPI-4?

Reply: Thanks for the valuable comment. We have compared the pigment arrangement in Lhcb9 of *Physcomitrium patens* (*Pp*) and those in *Symbiodinium* (*Sy*) PCPI-1, PCPI-3, and PCPI-4 which rotated by 180, 180, and 90 degrees, respectively (Fig. R1 below, also added to Supplementary Figure 23). In Lhcb9, the red Chl pair 603/609, which corresponds to the Chl pair 305/306 in PCPIs, is faced outward and is sandwiched by two carotenoids, L2 and L3. The Chl pair 305/306 and the carotenoid L2 are present in PCPI-1/3/4, whereas the carotenoid L3 exists only in PCPI-1. Therefore, the pigment arrangement of PCPI-1 is similar with that of Lhcb9, which is assumed to play a role in energy quenching in addition to light harvesting. We have added relevant descriptions in RESULTS AND DISCUSSION (lines 337-344) of the revised manuscript.

Fig. R1. Comparison of the arrangement of the Chl pair 603/609 and carotenoids L2/L3 in Lhcb9 of moss *Physcomitrium patens* (*Pp*) with those in *Symbiodinium* (*Sy*) PCPI-1, PCPI-3, and PCPI-4. The red Chl pair 603/609 (corresponding to the Chl pair 305/306 in PCPI-1/3/4) and the two carotenoids L2/L3 of Lhcb9 are labelled.

8. About the material: Does the structure of the complexes change when algae transition from a free-living to a symbiotic lifestyle?

Reply: Thanks for the great comment. We have now specified that the complex was obtained from *Symbiodinium sp.* in a free-living state in the revised manuscript (RESULTS AND DISCUSSION). Previous studies reported the detection of numerous differentially expressed genes (DEGs) between the symbiotic and free-living states of a coral symbiont genus (Yuyama *et al.*, *Microorganisms*, 2021, 9(8):1560). The results documented that the expression of genes related to photosynthesis was prone to increase during endosymbiosis, suggesting an enhancement in photosynthetic activity in algae when they are in endosymbiosis with corals. This finding is consistent with previous work on endosymbiosis in *Symbiodiniaceae*, which indicated that the symbiotic environment promotes photosynthesis in the symbiotic algae (Barott *et al.*, *PNAS*, 2014, 112: 607-612). Our study characterised the structure of photosystem I of *Symbiodinium* in its free-living state. Given that photosynthesis in symbiotic algae is enhanced during endosymbiosis, the structures of their photosystems may undergo structural changes. The conformational changes of photosystems as the symbiotic algae transition from a free-living to a symbiotic lifestyle remain a mystery and merits further investigations.

9. Supplementary Table. 3/ PsaR also exist in red algal PSI-LHC (You et.al, Nature, 2023), the author should revise it in this table.

Reply: Thanks for pointing this out. We have updated Supplementary Table 3 with additional information.

REVIEWER COMMENTS

Reviewer #1 (Remarks to the Author):

The authors have addressed all my comments appropriately and I am now happy to recommend the manuscript for publication.

Reviewer #2 (Remarks to the Author):

The information of the isolate of Symbiodinium is missing: where was it isolated? How was it classified as Symbiodinium given the many genera that have been established for coral endosymbionts? How was the isolate determined to be monospecific? Where is the isolate? The strain must be deposited in a public algal culture collection (e.g. NCMA) for public access for reproducing the work.

Given the known substantial genetic divergence between genotypes and species, the accurate species identification information is crucial. Proof of monospecific nature of the isolate is essential. The availability of the isolate to researchers is also imperative.

Line 464: what wavelength of light was used in SP? Same question for subsequent steps of manipulation.

Line 466: how was the “fully reduced with great rapidity” determined? Same question for lines 467-468.

Line 507: As PsaT and PsaU have not been discovered in any photosynthetic organisms, how come the diatom homologs for use here as a reference?

Lines 509-510: a question was raised in the first round of review, and it seems to stay unaddressed here: how can the “best match” be defined without knowing the sequences in the first place?

Lines 513-514: do you mean assign both putative diadinoxanthin and putative diatoxanthin as diadinoxanthin?

Line 519: with what criteria to terminate the iteration?

PsaT and PsaU peptide sequences should be made available for review now and for public access when the paper is published.

Given the multiple Symbiodiniaceae genomes currently available, PsaT and PsaU homologs should be identified and their potential evolutionary origin discussed.

I find it unnecessary, and potentially confusing, to name the PCP in PSI as PCPI in the paper (including figures), because the work does not involve PSII and PCPI gives incorrect impression as a different

structure than PCP. It is particularly confusing when PSI-PCPI is used, because it implies there is a PSI-PCPII etc.

In Figure 5c, P1, P2, P3, ...: do they stand for PCP1, PCP2, PCP3, ...? Explain it in the legend.

Summarizing the findings with a schematic depicting the organization of the PSI and light harvesting complex components would be very meaningful.

Related to the last comment, some comparative description of dinoflagellate PSI and diatom PSI would be more meaningful than discussing the Symbiodinium PSI in the context of coral symbiosis. This is because both dinoflagellates and diatoms are important primary producers in the ocean but they adapt to different light fields: diatoms are more flexible to fluctuating light condition than dinoflagellates; diatoms are more adapted to turbid water whereas dinoflagellates prefer clear water, generally speaking.

References: refs 20 and 21 deal with quite different questions and the citation for the same context is a bit out of place. Anyway, the major conclusion of ref 21 has been contended and does not stand any longer (see <https://doi.org/10.1078/1434461042650325>). The authors are recommended to refer to up-to-date mainstream literature on dinoflagellates.

Finally, as the manuscript introduces the work with the mention that PSI is important for photoprotection, some more discussion should be given as to how the observed PSI-PCP architecture confers a strong photoprotective capacity. How the binding to dtx and ddx help NPQ? Through electron flow?

Related to the last comment, in line 118-120: how does this running of Psal/J/L/M/R extended termini over the stromal surface of the PSI core provide a protection? Protection against light or what?

Reviewer #3 (Remarks to the Author):

The author has revise their manuscript and answered my questions well. I recommend the manuscript for publication.

Responses to Reviewers' Comments

Reviewer #1:

The authors have addressed all my comments appropriately and I am now happy to recommend the manuscript for publication.

Reply: We sincerely appreciate the reviewer for endorsing publication.

Reviewer #3:

The author has revise their manuscript and answered my questions well. I recommend the manuscript for publication.

Reply: We sincerely appreciate the reviewer for endorsing publication.

Reviewer #2:

1. The information of the isolate of *Symbiodinium* is missing: where was it isolated? How was it classified as *Symbiodinium* given the many genera that have been established for coral endosymbionts? How was the isolate determined to be monospecific? Where is the isolate? The strain must be deposited in a public algal culture collection (e.g. NCMA) for public access for reproducing the work.

Reply: Thanks for raising this point. The strain used in this study, *Symbiodinium* sp. GY-H50, was purchased from the commercial algae bank, Shanghai Guangyu Biological Technology Co., Ltd.. Therefore, we could not deposit it in a public algal culture collection. According to the information provided by the company (see the strain certificate below), the species was isolated by limiting dilution method from coral in Beibu Gulf (Guangxi province, China) and was monospecific. It was identified as *Symbiodinium* based on 18S ribosomal RNA gene sequencing. To verify the strain, we have also performed 18S ribosomal RNA and internal transcribed spacer (ITS) gene sequencing, confirming that the strain belongs to *Symbiodinium*. The ribosomal RNA gene sequences have been deposited to the National Center for Biotechnology Information database with the accession numbers PP191135 and PP191136. We have now added the specification and Supplementary Table 8 in the revised manuscript.

Certificate

This is to certify that the Marine Biotechnology Research Center, State Key Laboratory of Microbial Technology of Shandong University (Qingdao 266237, China) has purchased the algae, *Symbiodinium* sp. GY-H50, from the commercial algae bank, Shanghai Guangyu Biological Technology Co., Ltd. at September, 2020. This alga was isolated by limiting dilution method from coral in Beibu Gulf (Guangxi province, China), and was monospecific. The alga was identified as *Symbiodinium* by 18S ribosomal RNA gene sequencing.

Shanghai Guangyu Biological Technology Co., Ltd.
Building 4, No. 398 Xiangche Road
Chedun Town, Songjiang District
Shanghai, China
Jan. 21, 2024

Fig. R1. Certificate of the algal strain in this study

2. Line 464: what wavelength of light was used in SP? Same question for subsequent steps of manipulation.

Reply: The maximum intensity of the light for saturation pulse (SP) and actinic light (AL) was 625 nm. We have clarified this in METHODS.

3. Line 466: how was the “fully reduced with great rapidity” determined? Same question for lines 467-468.

Reply: A strong intensity of saturation pulse (SP) was used in this study ($10,000 \mu\text{M photons m}^{-2} \text{s}^{-1}$). The 200 ms SP releases a significant amount of photons in a short time and induces multiple charge separations in the PSII reaction center. The resulting electrons are sufficient to fully reduce the secondary electron acceptor, plastoquinone (PQ). SP also induces multiple charge separations in the PSI reaction center, resulting in complete oxidation of P700 (Pm) (Fig. R2). After SP was turned off, the oxidized P700 was fully reduced with great rapidity by electrons from PSII through PQ, cytochrome (Cyt) b_6 , and plastocyanin (PC) (Fig. R2).

Fig. R2. The curve of P700 redox kinetic.

4. Line 507: As *PsaT* and *PsaU* have not been discovered in any photosynthetic organisms, how come the diatom homologs for use here as a reference?

Reply: We would like to clarify that we only used diatom homologs (FCPIs) for determining the PCPI sequences, instead of *PsaT* and *PsaU*. To avoid the confusion, we have revised the corresponding descriptions in METHODS (Page 19). “The potential sequences of PCPIs were searched from the transcriptome by blast using their homologous sequences from diatom (FCPIs). The potential sequences of *PsaT* and *PsaU* were searched from the transcriptome by blast using the estimated sequences of *PsaT* and *PsaU* based on their cryo-EM density maps (Supplementary Fig. 3). As the side chains of residues could not be identified precisely based on the cryo-EM density map at the current resolution, several sequences of *PsaT* and *PsaU* were estimated and each sequence is about 20 residues. Then, each of the potential sequences of *PsaT*, *PsaU*, and PCPIs was manually fitted into the density map, and amino acids with distinguishable side chains (such as phenylalanine, tyrosine, tryptophan, arginine, and glycine) were used to identify the proper sequence from the potential sequences (Supplementary Fig. 3).”.

5. Lines 509-510: a question was raised in the first round of review, and it seems to stay unaddressed here: how can the “best match” be defined without knowing the sequences in the first place?

Reply: Please see our responses for Q4 above. We have revised the corresponding descriptions and provided detailed clarification in METHODS (Page 19).

6. Lines 513-514: do you mean assign both putative diadinoxanthin and putative diatoxanthin as diadinoxanthin?

Reply: We have revised the description in METHODS (Page 19). Given the current resolution of the density map, it's impossible to differentiate between diadinoxanthin and diatoxanthin. Therefore, the binding sites for both the putative diadinoxanthin and putative diatoxanthin were assigned as diadinoxanthin.

7. Line 519: with what criteria to terminate the iteration?

Reply: During the model refinement, we evaluated the geometries of the structural model after each round of manual correction and Phenix refinement using the wwPDB Validation System and the validation program of Phenix. After several rounds of refinement, the model had no outliers in bond length, bond angle, chirality, and planarity. As shown in the validation report, the Ramachandran outliers and sidechain outliers were 0.04% and 0.12%, respectively, which were better than most of the EM structures. Based on these parameters, we believe that the structural model in this study is qualified for publication.

8. *PsaT* and *PsaU* peptide sequences should be made available for review now and for public access when the paper is published.

Reply: Thanks for the suggestion. The ribosomal RNA gene sequences of PsaT and PsaU have been deposited to the National Center for Biotechnology Information database with the accession numbers of PP196340 and PP196339, respectively. We have clarified this in the main text and Data Availability.

9. Given the multiple Symbiodiniaceae genomes currently available, PsaT and PsaU homologs should be identified and their potential evolutionary origin discussed.

Reply: Thanks for the suggestions. We have identified the homologous sequences of PsaT and PsaU. As stated in the manuscript, the homologous sequences are exclusive in *Symbiodiniaceae* and are only present in various species of *Symbiodiniaceae*. This suggests that PsaT and PsaU may play an important role in the adaptation and survival of *Symbiodiniaceae* in varying environmental conditions during evolution. We have added the phylogenetic analysis in Fig. R3 below and Supplementary Fig. 9j-k in the revised manuscript. Their potential evolutionary origin and physiological functions in the PSI-PCPI supercomplex merit further investigation.

Fig. R3. Phylogenetic tree of PsaT, PsaU and their homologous sequences. a, Phylogenetic tree of PsaT and its homologous sequences. **b,** Phylogenetic tree of PsaU and its homologous sequences. The neighbor-joining tree was based on amino acid sequences of LHCI. The phylogenetic trees were built with the Poisson model, and a bootstrap test (1,000 replicates) was conducted.

10. I find it unnecessary, and potentially confusing, to name the PCP in PSI as PCPI in the paper (including figures), because the work does not involve PSII and PCPI gives incorrect impression as a different structure than PCP. It is particularly confusing when PSI-PCPI is used, because it implies there is a PSI-PCPII etc.

Reply: We would like to emphasise that this is a general method approach for naming LHC that associate with PSI and PSII in photoautotrophic organisms. In green algae, moss, and land plants, the LHC complexes associated with PSI were specifically named as LHCI and those associated with PSII were termed as LHCII (Qin *et al.*, *Science*, 2015, 348:989-995; Qin *et al.*, *Nat Plants*, 2019, 5:263-272; Gorski *et al.*, *Nat Plants*, 2022, 8:307-316; Wei *et al.*, *Nature*, 2016, 534:69-74; Sheng *et al.*, *Nature Plants*, 2019, 5:1320-1330). In diatom, fucoxanthin-chlorophyll *a/c* proteins (FCPs) function as LHC, and the FCPs associated with PSI were named as FCPI, and those associated with PSII were termed FCPII (Xu *et al.*, *Nat Commun*, 2020, 11:5081; Pi *et al.*, *Science*, 2019, 365:eaax4406). Additionally, in our previous study on the structure of cryptophyte PSI-LHCI, we named ACPI (alloxanthin-chlorophyll *a/c* protein) as the LHCI associated with PSI (Zhao *et al.*, *Plant Cell*, 2023, 35:2449-2463). To keep the consistency in the field, we followed the conventional nomenclature and named the PCP in PSI as PCPI. We have specified this in the revised manuscript (Page 4).

11. In Figure 5c, P1, P2, P3, ...: do they stand for PCPI, PCP2, PCP3, ...? Explain it in the legend.

Reply: We have added the clarification in the Figure 5 legend as suggested by the reviewer.

12. Summarizing the findings with a schematic depicting the organization of the PSI and light harvesting complex components would be very meaningful.

Reply: Thanks for the suggestion. We have added the schematic models in Supplementary Fig. 14f to illustrate the organizations of the PSI core and LHC components in *Symbiodinium* PSI-PCPI, red algae PSI-LHCR,

cryptophyte PSI–ACPI, and diatom PSI–FCPI. We have added relevant discussion in the revised manuscript (Page 9).

13. Related to the last comment, some comparative description of dinoflagellate PSI and diatom PSI would be more meaningful than discussing the *Symbiodinium* PSI in the context of coral symbiosis. This is because both dinoflagellates and diatoms are important primary producers in the ocean but they adapt to different light fields: diatoms are more flexible to fluctuating light condition than dinoflagellates; diatoms are more adapted to turbid water whereas dinoflagellates prefer clear water, generally speaking.

Reply: Thanks for the great suggestions. In this study, we compared the structures of the PSI core subunits and LHCs, the arrangements of LHCs and pigments, and the EET pathways in *Symbiodinium* PSI–PCPI with those of diatom PSI–FCPI, highlighting their remarkable differences. These structural variations between PSI–PCPI and PSI–FCPI could be attributed to the adaptations of dinoflagellates and diatoms to their respective environmental conditions. We have added relevant discussion in the revised manuscript (Page 14-15).

14. References: refs 20 and 21 deal with quite different questions and the citation for the same context is a bit out of place. Anyway, the major conclusion of ref 21 has been contended and does not stand any longer (see <https://doi.org/10.1078/1434461042650325>). The authors are recommended to refer to up-to-date mainstream literature on dinoflagellates.

Reply: Thanks for pointing this out. We have updated the literature on dinoflagellates.

15. Finally, as the manuscript introduces the work with the mention that PSI is important for photoprotection, some more discussion should be given as to how the observed PSI-PCP architecture confers a strong photoprotective capacity. How the binding to dtx and ddx help NPQ? Through electron flow?

Reply: Carotenoids play dual roles in the processes of excitation energy transfer: transferring excitation energy to Chls and quenching excessive excitation energy from Chls for photoprotection. The PCPI-associated carotenoids have close interactions with Chls located within or around PCPIs (Supplementary Fig. 21k). This suggests that there is a highly efficient energy transfer between these pigment molecules. In addition, it is worth noting that *Symbiodinium* PSI–PCPI may possess multiple distinct energy quenching sites that are formed by the presence of new Chls and new Cars (Supplementary Fig. 21k-21t). The new Cars identified in PSI–PCPI are close to Chls and are presumably responsible for dissipating excessive excitation energy from Chls (Supplementary Fig. 21k-21q). The newly identified Chls in PSI–PCPI exhibit strong associations with Cars and have the potential to transfer excessive excitation energy to these Cars (Supplementary Fig. 21q-21t). We have added additional panels (k-t) in Supplementary Fig. 21 to illustrate the potential energy quenching sites and have added relevant discussion in the revised manuscript (Page 10-11).

The Ddx-Dtx cycle converts the monoepoxide carotenoid, Ddx, into the de-epoxide form, Dtx, when exposed to intense light (Yamamoto *et al.*, *Meth Enzymol*, 1985, 110: 303-312). Dtx dissipates excessive excitation energy through fluorescence quenching and heat dissipation, resulting in significant NPQ (Demers *et al.*, *Mar Ecol Progr Ser*, 1991, 76:185-193; Olaizola and Yamamoto, *Journal of Phycology*, 1994, 30:606-612; Lavaud *et al.*, *Plant Physiol*, 2002, 129:1398-1406; Goss *et al.*, *J Plant Physiol*, 2006, 163:1008-1021).

16. Related to the last comment, in line 118-120: how does this running of PsaI/J/L/M/R extended termini over the stromal surface of the PSI core provide a protection? Protection against light or what?

Reply: The extended termini of PsaI/J/L/M/R are located on the stromal surface of the PSI reaction centre, resulting in reduced interactions between the reaction centre and molecules in the stroma. This may mitigate the damage of reactive oxygen species to the reaction centre and protect the pigments in the reaction centre under high light. In addition, these extended termini form close interactions with the PSI core subunits and PCPIs, which may enhance the structural stability of PSI–PCPI under stress conditions. However, the function of these extended termini remains a mystery and merits further investigation. We have added relevant discussion in the revised manuscript (Page 5).

REVIEWERS' COMMENTS

Reviewer #2 (Remarks to the Author):

I am happy with the new revision, and I view the manuscript as being acceptable now.